# α-OCC: Uncertainty-Aware Camera-based 3D Semantic Occupancy Prediction

**Sanbao Su**                                                    *sanbao.su@uconn.edu*
*School of Computing*
*University of Connecticut*

**Nuo Chen**                                                        *nc3144@nyu.edu*
*Tandon School of Engineering*
*New York University*

**Chenchen Lin**                                            *chenchen.lin@uconn.edu*
*School of Computing*
*University of Connecticut*

**Felix Juefei-Xu**                                            *juefei.xu@nyu.edu*
*New York University*

**Chen Feng**                                                      *cfeng@nyu.edu*
*Tandon School of Engineering*
*New York University*

**Fei Miao**                                                    *fei.miao@uconn.edu*
*School of Computing*
*University of Connecticut*

**Reviewed on OpenReview:** *https://openreview.net/forum?id=bUv25gBLlV*

## Abstract

Comprehending 3D scenes is paramount for tasks such as planning and mapping for autonomous vehicles and robotics. Camera-based 3D Semantic Occupancy Prediction (OCC) aims to infer scene geometry and semantics from limited observations. While it has gained popularity due to affordability and rich visual cues, existing methods often neglect the inherent uncertainty in models. To address this, we propose an uncertainty-aware OCC method (α-OCC). We first introduce Depth-UP, an uncertainty propagation framework that improves geometry completion by up to 11.58% and semantic segmentation by up to 12.95% across various OCC models. For uncertainty quantification (UQ), we propose the hierarchical conformal prediction (HCP) method, effectively handling the high-level class imbalance in OCC datasets. On the geometry level, the novel KL-based score function significantly improves the occupied recall (45%) of safety-critical classes with minimal performance overhead (3.4% reduction). On UQ, our HCP achieves smaller prediction set sizes while maintaining the defined coverage guarantee. Compared with baselines, it reduces up to 90% set size, with 18% further reduction when integrated with Depth-UP. Our contributions advance OCC accuracy and robustness, marking a noteworthy step forward in autonomous perception systems. Our code is public on https://coperception.github.io/alpha-OCC/.

## 1 Introduction

Achieving a comprehensive understanding of 3D scenes is crucial for downstream tasks such as planning in autonomous vehicles (AVs) and robotics (Wang & Huang (2021)). 3D Semantic Occupancy Prediction (OCC)

emerges as a solution that jointly infers the geometry completion and semantic segmentation (Song et al. (2017); Hu et al. (2023)). OCC approaches typically are categorized based on the sensor input: LiDAR-based OCC and camera-based OCC. While LiDAR offers precise depth information (Roldao et al. (2020); Cheng et al. (2021)), they are costly and less portable. Conversely, camera-based OCCs, with their affordability and ability to capture rich visual cues, have gained significant attention (Cao & De Charette (2022); Tian et al. (2024)). For them, depth estimation is essential for the accurate 3D reconstruction of scenes. However, existing methods frequently overlook real-world depth estimation errors (Poggi et al. (2020)). Moreover, effectively leveraging propagated depth uncertainty and rigorously quantifying OCC uncertainty, particularly in class-imbalanced datasets, remains a challenging and unexplored problem. Throughout this paper, OCC refers to camera-based OCC unless stated otherwise, which is the focus of our work.

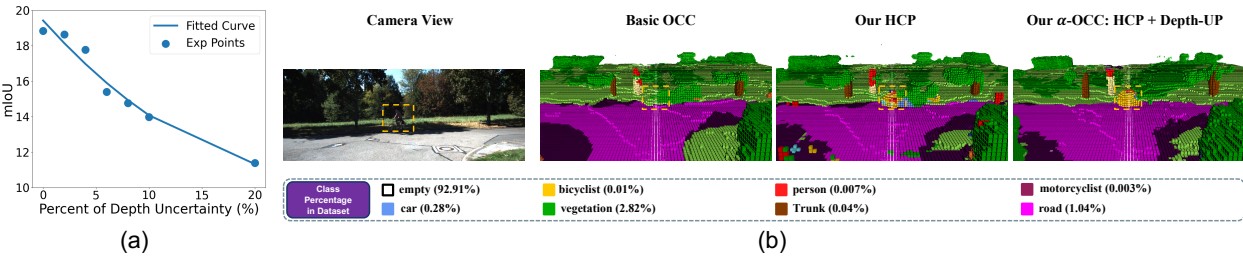

Figure 1: (a): As the percentage of depth uncertainty increases, the accuracy (mIoU↑) of OCC decreases significantly. (b): High class imbalance on OCC. The percentage next to each class is its percentage in the SemanticKITTI dataset. Since the safety-critical class "bicyclist" only occupied 0.01%, the trained OCC model fails to detect the bicyclist in front, leading to a crash. However, after quantifying the uncertainty and post-processing using our HCP, the crash is avoided, for our HCP improves the occupied recall of rare classes. When applying our Depth-UP and HCP together, safety is further enhanced as the bicyclist is more accurately identified. In contrast, using only HCP often assigns the highest probability to the car (blue) for many bicyclist voxels. Due to visualization limits, each occupied voxel shows the most probable nonempty class in the predicted set from HCP.

We explain the importance of considering depth uncertainty propagation and OCC uncertainty quantification in Fig. 1. In Fig. 1(a), perturbing the ground-truth depth values by factors of $(1 + \beta)$ ($\forall \beta \in \{0\%, 2\%, 4\%, 6\%, 8\%, 10\%, 20\%\}$), simulates real-world depth estimation errors. Uncertainties of depth estimation significantly reduce the performance of OCCs. To address this, we propose a flexible uncertainty propagation framework (Depth-UP) from depth models to improve the performance of a variety of OCC models.

The OCC datasets exhibit a high class imbalance, with empty voxels comprising a significant proportion (92.91% for SemanticKITTI Behley et al. (2019)), as illustrated in the dotted box of Fig. 1(b). Bicyclist and person voxels, crucial for safety, only occupy 0.01% and 0.007% of the dataset. Consequently, networks trained on such imbalanced data, coupled with the maximum posterior classification, often overlook rare classes, leading to reduced accuracy and recall (Tian et al. (2020)). This poses a significant risk in safety-critical applications like AVs, where detecting rare-class occupied voxels is essential for preventing collisions (Chan et al. (2019)). As in Fig. 1(b), a standard OCC model fails to detect the bicyclist due to the extreme rarity of the bicyclist class in the dataset. To address this, we propose a hierarchical conformal prediction (HCP) method that enhances occupied recall for rare classes in geometry completion while generating prediction sets with class coverage guarantees for semantic segmentation. By quantifying uncertainty (prediction set) through HCP, the OCC model successfully identifies rare bicyclist voxels, reducing the risk of collisions. Furthermore, our complete $\alpha$-OCC framework integrates both HCP and Depth-UP, further refines bicyclist detection, significantly improving safety.

Extensive experiments on three OCC models (VoxFormer Li et al. (2023b), OccFormer Zhang et al. (2023) and CGFormer Yu et al. (2024)) and two datasets (SemanticKITTI Behley et al. (2019) and KITTI360 Li et al. (2023a)) show that our Depth-UP improves geometry completion by 11.58% and semantic segmentation by 12.95%. Our HCP enhances person-class geometry prediction by 45% with only 3.4% IoU overhead,

improving prediction on rare safety-critical classes, such as persons, thereby reducing potential risks for AVs. Compared with baselines, HCP reduces set size by 90% and coverage gap by 64%, an additional 18% set size reduction when combined with Depth-UP. These results show the significant accuracy and uncertainty quantification improvements of our $\alpha$-OCC.

Our contributions are unified under the proposed $\alpha$-OCC, an uncertainty-aware camera-based 3D semantic occupancy prediction framework. This approach recognizes the OCC problem from a fresh uncertainty perspective and consists of two major components: 1) To the best of our knowledge, we are the first to propose the uncertainty propagation framework Depth-UP to improve OCC performance. Our approach leverages uncertainty quantified through direct modeling to improve both geometry completion and semantic segmentation, resulting in substantial performance gains across common OCC models. 2) To solve the high-level class imbalance challenge on OCC, resulting in biased prediction and low recall for rare classes, we propose the HCP. On geometry completion, a novel KL-based score function is proposed to improve the occupied recall of safety-critical classes with little performance overhead. For UQ, we achieve a smaller prediction set size under the defined class coverage guarantee. Overall, our $\alpha$-OCC shows that uncertainty is an integral and vital part of OCC tasks. Integrating Depth-UP (propagating depth uncertainty to OCC) and HCP (quantifying OCC uncertainty) enhances both accuracy and uncertainty of OCC models.

## 2 Related Work

**Uncertainty Quantification and Propagation.** Uncertainty quantification (UQ) holds paramount importance in ensuring the safety and reliability of autonomous systems such as robots (Jasour & Williams (2019)) and AVs (Meyer & Thakurdesai (2020); Su et al. (2025)). Moreover, UQ for perception can significantly enhance the planning and control processes of them (Xu et al. (2014); He et al. (2023)). Different types of UQ methods have been proposed. Monte-Carlo dropout (Miller et al. (2018)) and deep ensemble (Lakshminarayanan et al. (2017)) methods require multiple runs of inference, which makes them infeasible for real-time applications. In contrast, direct modeling (Feng et al. (2021)) estimates uncertainty in one pass, used to estimate the depth uncertainty in our work.

While uncertainty has been explored in 3D tasks, our focus differs. Eldesokey et al. (2020) improved depth completion with uncertainty by normalized convolutional neural networks. Cao et al. (2024) managed LiDAR-based OCC uncertainty with deep ensembles, increasing computational complexity. Although uncertainty propagation (UP) from depth to 3D object detection has improved accuracy (Lu et al. (2021); Wang et al. (2023)), no prior work has addressed UP from depth to OCCs. To bridge this gap, we propose Depth-UP, a direct modeling-based UP for OCCs.

Conformal prediction (CP) provides statistically guaranteed uncertainty sets for model predictions (Angelopoulos & Bates (2021); Su et al. (2024)). However, its application to class-imbalanced tasks is limited. Rare and safety-critical classes (e.g., person) remain challenging for OCC models. To address this, we propose a hierarchical conformal prediction method for uncertainty quantification in highly class-imbalanced OCCs. Related works on class imbalance are introduced in Appendix A.1. And related works on OCC are introduced in Appendix A.2.

## 3 Method

We propose $\alpha$-OCC, an uncertainty-aware OCC method integrating Depth-UP for uncertainty propagation and HCP for uncertainty quantification. Figure 2 presents the whole methodology overview and the structure of our Depth-UP. Figure 3 details our HCP. The major novelties are: *(1)* Depth-UP quantifies the uncertainty of depth estimation by direct modeling (DM) and then propagates it through probabilistic geometry projection (for geometry completion) and depth feature extraction (for semantic segmentation). *(2)* HCP calibrates OCC probability outputs. First, it predicts the voxels' occupied state by the quantile on the novel KL-based score function as Eq. 2, which can improve the occupied recall of rare safety-critical classes. Then it generates prediction sets for predicted occupied voxels, achieving a better coverage guarantee and smaller prediction set sizes.

**Preliminary:** OCC predicts a dense semantic scene within a defined volume in front of the vehicle solely from RGB images (Cao & De Charette (2022)). Specifically, with an input image $\mathbf{X} \in \mathbb{R}^{3 \times H \times W}$, an OCC model first extracts 2D image features $\mathbf{F}_I$ using backbone networks like ResNet (He et al. (2016)) and estimates

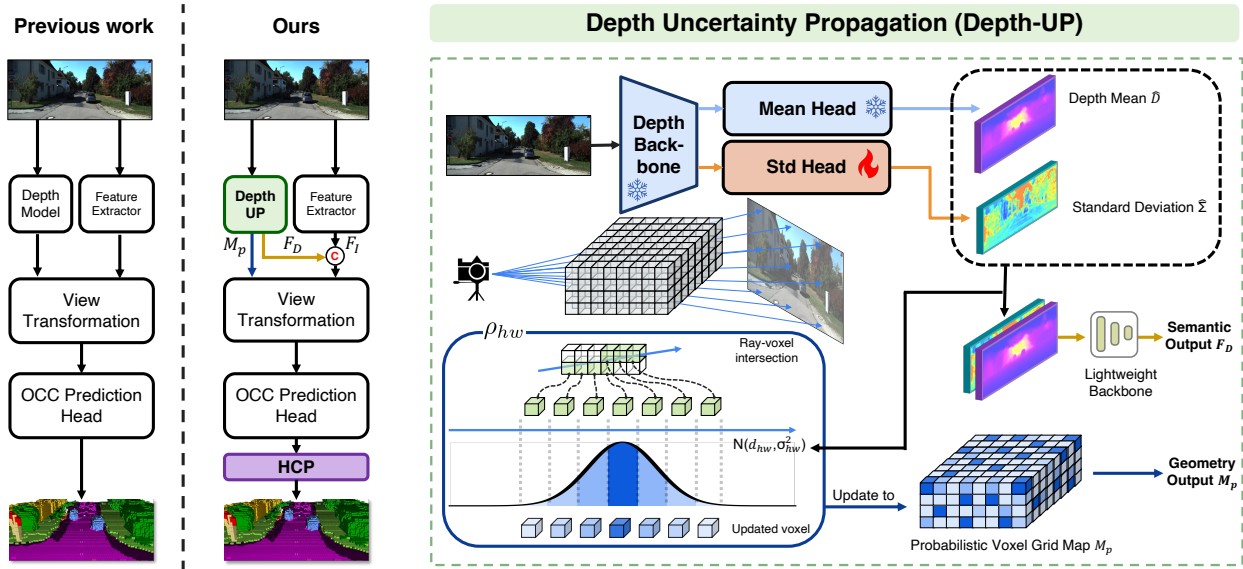

Figure 2: Overview of our $\alpha$-OCC method. The non-black colors highlight the novelties and important techniques in our method. **C** denotes the concatenation of the depth feature $\mathbf{F}_D$ and image feature $\mathbf{F}_I$. In the Depth-UP part, we calculate the uncertainty of depth estimation through direct modeling. For depth model retraining, we only train the additional standard deviation head while keeping the rest of the model frozen. Then we propagate it through depth feature extraction (for semantic segmentation) and building a probabilistic voxel grid map $M_p$ by probabilistic geometry projection (for geometry completion). Each element of $M_p$ is the occupied probability of the corresponding voxel, computed by considering the depth distribution of all rays across the voxel.

the depth value $\hat{\mathbf{D}} \in \mathbb{R}^{H \times W}$ for each pixel (e.g. monocular depth estimation Bhat et al. (2021) or stereo depth estimation Shamsafar et al. (2022)). Then, a probability voxel grid $\hat{\mathbf{Y}} \in [0,1]^{M \times U \times V \times D}$ is generated based on $\mathbf{F}_I$ and $\hat{\mathbf{D}}$, assigning each voxel to the class with the highest probability. Each voxel within the grid is categorized as either empty or occupied by a specific semantic class. The ground truth voxel grid is denoted as $\mathbf{Y}$, $H$ and $W$ is the height and width of the input image, $U$, $V$ and $D$ represent the height, width, and length of the voxel grid, $M$ denotes the total number of relevant classes (including the empty class), respectively.

## 3.1 Uncertainty Propagation Framework (Depth-UP)

In OCC methods, depth models facilitate the projection from 2D to 3D space, primarily focusing on geometric aspects. Nonetheless, these approaches often overlook the inherent depth uncertainty. Recognizing the potential to enhance OCC performance by utilizing this uncertainty, we introduce Depth-UP framework, which quantifies the uncertainty of depth models through a direct modeling method and propogate this uncertainty into both geometry completion and semantic segmentation to improve the final performance of various OCC models.

**Direct Modeling (DM).** Depth-UP includes a DM technique (Su et al. (2023); Feng et al. (2021)) to estimate the standard deviation associated with the estimated depth value of each pixel, with minimal computational overhead. An additional regression head, with a comparable structure as the original regression head for $\hat{\mathbf{D}}$, is tailored to predict the standard deviation $\hat{\boldsymbol{\Sigma}}$. Subsequently, this head is retrained based on the original depth model, with all parameters of the original depth model (including the original regression head) being frozen. Assume the estimated depth value is represented as a univariate Gaussian distribution, and the ground truth depth follows a Dirac delta function (Arfken et al. (2011)). For the retraining process, we define the regression loss function as the KL-divergence between the estimated distribution

and the ground truth distribution, where $\mathbf{D} \in \mathbb{R}^{H \times W}$ is the ground truth depth matrix for the image: $\mathcal{L}_{KL}(\mathbf{D}, \hat{\mathbf{D}}, \hat{\mathbf{\Sigma}}) = \frac{1}{HW} \sum_{h=1}^{H} \sum_{w=1}^{W} \frac{(d_{hw} - \hat{d}_{hw})^2}{2\hat{\sigma}_{hw}^2} + \log |\hat{\sigma}_{hw}|$.

**Propagation on Geometry Completion (PGC).** Depth information is used to generate the 3D voxels on geometry in OCC. There are two key challenges: lens distortion during geometric transformations and occupied probability estimation for each voxel. Lens distortion is a deviation from the ideal image formation by a lens, resulting in a distorted image (Zhang (2000)). Existing OCC models, such as VoxFormer (Li et al. (2023b)), solve the lens distortion by projecting depth into a 3D point cloud, and then generating the binary voxel grid map $\mathbf{M}_b \in \{0, 1\}^{U \times V \times D}$, where each voxel is marked as 1 if occupied by at least one point. However, they ignore the uncertainty of depth. Here we propagate the depth uncertainty into the geometry of OCC to solve the above two challenges.

Our Depth-UP generates a **probabilistic voxel grid map $\mathbf{M}_p \in [0, 1]^{U \times V \times D}$** that considers lens distortion and depth uncertainty, with $\{\hat{\mathbf{D}}, \hat{\mathbf{\Sigma}}\}$ from the depth model. For pixel $(h, w)$ with estimated depth mean $\hat{d}_{hw}$, we project it into point $(x, y, z)$ in 3D space: $x = \frac{(h - c_h) \times z}{f_u}, y = \frac{(w - c_w) \times z}{f_v}, z = \hat{d}_{hw}$, where $(c_u, c_v)$ is the camera center and $f_u$ and $f_v$ are the horizontal and vertical focal length.

When estimated depth follows a univariate Gaussian distribution, a point's exact location along its camera ray is uncertain. Instead, we estimate the probability of one voxel $(u, v, d)$ being occupied by points. Due to the density of visual information, a single voxel may correspond to multiple pixels, which means a voxel can be passed by multiple rays. We denote this set of rays as $\Psi_{uvd}$, and a single ray within this set as $\rho_{hw}$, corresponding to pixel $(h, w)$. When a ray $\rho_{hw}$ passes through a voxel, it has two crosspoints: $z_s$ where the ray enters the voxel, and $z_e$ where the ray exits the voxel. By cumulating the probability of the ray inside the voxel using the probability density function, we obtain the probability of voxel $(u, v, d)$ being occupied by points:

$$\mathbf{M}_p(u, v, d) = \min \left( 1, \sum_{\rho_{hw} \in \Psi_{uvd}} \int_{z_s}^{z_e} \mathcal{N}(z | \hat{d}_{hw}, \hat{\sigma}_{hw}^2) dz \right). \tag{1}$$

The original binary voxel grid map is replaced by the probabilistic voxel grid map $\mathbf{M}_p \in [0, 1]^{U \times V \times D}$ to propagate the depth uncertainty into the geometry completion of OCC.

**Propagation on Semantic Segmentation (PSS).** The extraction of 2D features $\mathbf{F}_I$ from the input image has been a cornerstone for OCC to encapsulate semantic information. However, utilizing depth uncertainty on the semantic features is ignored. Through an additional lightweight backbone, such as ResNet-18 backbone (He et al. (2016)), we extract depth features $\mathbf{F}_D$ from the concatenated depth mean and standard deviation $\{\hat{\mathbf{D}}, \hat{\mathbf{\Sigma}}\}$. These newly acquired depth features are then seamlessly integrated with the original 2D image features, constituting a novel set of input features $\{\mathbf{F}_I, \mathbf{F}_D\}$ as shown in Figure 2. This integration strategy capitalizes on the extensive insights gained from prior depth predictions, enhancing the OCC performance with enhanced semantic understanding.

## 3.2 Hierarchical Conformal Prediction (HCP)

### 3.2.1 Preliminary

**Standard Conformal Prediction.** For classification, conformal prediction (CP, Angelopoulos & Bates (2021); Ding et al. (2024)) is a statistical method to post-process any models by producing the set of predictions with theoretically guaranteed marginal coverage of the correct class. With $M$ classes, considering the calibration data $(\mathbf{X}_1, \mathbf{Y}_1), ..., (\mathbf{X}_N, \mathbf{Y}_N)$ with N data points that are never seen during training, the standard CP (SCP) includes the following steps: *(1)* Define the score function $s(\mathbf{X}, y) \in \mathbb{R}$ (Smaller scores indicate better agreement between $\mathbf{X}$ and $y$). The score function is a vital component of CP. A typical score function of a classifier $f$ is $s(\mathbf{X}, y) = 1 - f(\mathbf{X})_y$, where $f(\mathbf{X})_y$ represents the $y^{th}$ softmax output of $f(\mathbf{X})$. *(2)* Compute $q$ as the $\frac{\lceil (N+1)(1-\alpha) \rceil}{N}$ quantile of the calibration scores, where $\alpha \in [0, 1]$ is a user-chosen error rate. *(3)* Use this quantile to form the prediction set $\mathcal{C}(\mathbf{X}_{test}) \subset \{1, ..., M\}$ for one new example $\mathbf{X}_{test}$ (from the same distribution of the calibration data): $\mathcal{C}(\mathbf{X}_{test}) = \{y : s(\mathbf{X}_{test}, y) \leq q\}$. The SCP provides a coverage guarantee that $\mathbb{P}(\mathbf{Y}_{test} \in \mathcal{C}(\mathbf{X}_{test})) \geq 1 - \alpha$ which has been proved in Angelopoulos & Bates (2021).

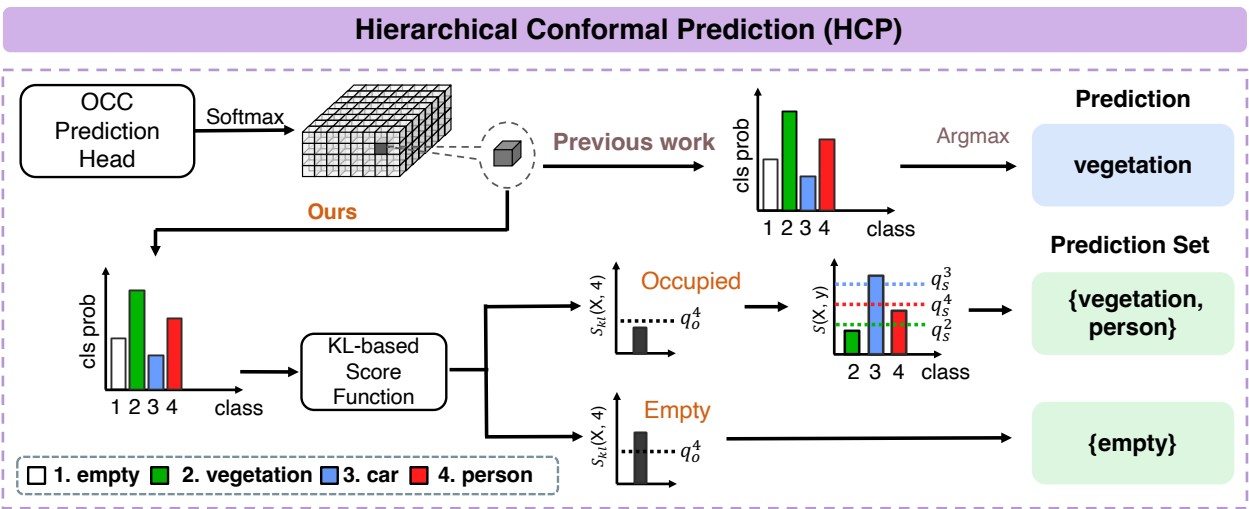

Figure 3: Overview of our HCP method. We predict voxels' occupied state by the quantile on the KL-based score as Eq. 2, which can improve occupied recall of rare classes, and then only generate prediction sets for these predicted occupied voxels. The occupied quantile $q_o^y$ and semantic quantile $q_s^y$ are computed during the calibration step of HCP.

**Class-Conditional Conformal Prediction.** The SCP achieves the marginal guarantee but may neglect the coverage of some classes, especially on class-imbalanced datasets (Angelopoulos & Bates (2021)). Class-Conditional Conformal Prediction (CCCP) targets class-balanced coverage under the user-chosen class error rate $\alpha^y$, $\forall y \in \{1, ..., M\}$: $\mathbb{P}(\mathbf{Y}_{test} \in \mathcal{C}(\mathbf{X}_{test})|\mathbf{Y}_{test} = y) \geq 1 - \alpha^y$. Every class $y$ has at least $1 - \alpha^y$ probability of being included in the prediction set when the label is $y$. Hence, the prediction sets are with coverage guarantee to all classes, even the rare ones.

### 3.2.2 Our Hierarchical Conformal Prediction

Current CP does not consider the hierarchical structure of classification, such as the geometry completion and semantic segmentation in OCCs. And it cannot achieve good coverage for very rare and safety-critical classes. Here we propose a novel HCP to address these challenges, as shown in Figure 3. The detailed algorithm is shown in Appendix A.3.

**Geometric Level.** On the geometric level, it is important and safety-critical to guarantee the occupied recall of some sensitive classes, such as the person and bicyclist for AVs. Hence, we define the occupied coverage for the specific safety-critical class $y$ as: $\mathbb{P}(o = T|\mathbf{Y}_{test} = y) \geq 1 - \alpha_o^y$, where $o = T$ means the occupancy state is true. The probability of the voxels with label $y$ are predicted as occupied is guaranteed to be no smaller than $1 - \alpha_o^y$. The empty class is $y = 1$ and occupied classes are $y \in \{2, ..., M\}$. To achieve the above guarantee under the high class-imbalanced dataset, we propose a novel score function based on the KL divergence. Here we define the ground-truth distribution for occupancy as $\mathbf{O} = \{\varepsilon, 1, ..., 1\}^M$, where $\varepsilon$ is the minimum value for the empty class to avoid the divide-by-zero problem. With the output softmax probability $f(\mathbf{X}) = \{p_1, p_2, ..., p_M\}$ from the model $f$, we define the KL-based score function for $y \in \mathcal{Y}_r$:

$$s_{kl}(\mathbf{X}, y) = D_{kl}(f(\mathbf{X})||\mathbf{O}) = p_1 \log(\frac{p_1}{\varepsilon}) + \sum_{i=2}^{M} p_i \log(p_i), \tag{2}$$

where $\mathcal{Y}_r$ is the considered rare class set. The quantile $q_o^y$ is computed as the $\frac{\lceil (N_y+1)(1-\alpha_o^y)\rceil}{N_y}$ quantile of the score $s_{kl}(\mathbf{X}, y)$ on $\Upsilon^y$, where $\Upsilon^y$ is the subset of the calibration dataset with $\mathbf{Y} = y$ and $N_y = |\Upsilon^y|$. Then we predict the voxel $\mathbf{X}_{test}$ as occupied if $\exists y \in \mathcal{Y}_r, s_{kl}(\mathbf{X}_{test}, y) \leq q_o^y$.

**Semantic Level.** On the semantic level, we need to achieve the same class-balanced coverage under the geometric level coverage guarantee. For all voxels predicted as occupied in the previous step, we generate the

prediction set $\mathcal{C}(\mathbf{X}_{test}) \subset \{2, ..., M\}$ to satisfy the guarantee:

$$\mathbb{P}(\mathbf{Y}_{test} \in \mathcal{C}(\mathbf{X}_{test}) | \mathbf{Y}_{test} = y, o = T) \geq 1 - \alpha_s^y. \tag{3}$$

The score function here is $s(\mathbf{X}, y) = 1 - f(\mathbf{X})_y$. We compute the quantile $q_s^y$ for class $y$ as the $\frac{\lceil (N_{yo}+1)(1-\alpha) \rceil}{N_{yo}}$ quantile of the score on $\Upsilon_o^y$, where $\Upsilon_o^y$ is the subset of the calibration dataset that has label $y$ and are predicted as occupied on the geometric level of our HCP. $N_{yo} = |\Upsilon_o^y|$. The prediction set is generated as:

$$\mathcal{C}(\mathbf{X}_{test}) = \{y : s_{kl}(\mathbf{X}, y) \leq q_o^y \wedge s(\mathbf{X}, y) \leq q_s^y\}. \tag{4}$$

**Proposition 1.** *For a desired error rate $\alpha^y$, we select $\alpha_o^y$ and $\alpha_s^y$ as $1 - \alpha^y = (1 - \alpha_s^y)(1 - \alpha_o^y)$, then the prediction set generated as Eq. 4 satisfies $\mathbb{P}(\mathbf{Y}_{test} \in \mathcal{C}(\mathbf{X}_{test}) | \mathbf{Y}_{test} = y) \geq 1 - \alpha^y$.*

The proof is in Appendix A.4.

## 4 Experiments

**OCC Model.** We assess the effectiveness of our approach through comprehensive experiments on three different OCC models: VoxFormer (Li et al. (2023b)), OccFormer (Zhang et al. (2023)) and CGFormer (Yu et al. (2024)). A detailed introduction to these three models is in Appendix A.2. **Dataset.** We use SemanticKITTI (Behley et al. (2019), with 20 classes) and KITTI360 (Li et al. (2023a), with 19 classes). (more details in Appendix A.5). Detailed experiment settings are in Appendix A.6. Experiments on the Occ3D-nuScenes (Caesar et al. (2020)) dataset are in Appendix A.9.

### 4.1 Uncertainty Propagation Performance

Table 1: Performance evaluation of our Depth-UP on three OCC models with two datasets. Values in parentheses indicate the improvement of our Depth-UP compared with the baseline.

| Dataset | OCC | Method | IoU ↑ | Precision ↑ | Recall ↑ | mIoU ↑ |
|---|---|---|---|---|---|---|
| SemanticKITTI | VoxFormer | Base | 44.02 | 62.32 | 59.99 | 12.35 |
| | | Our | 45.85 (+1.83) | 63.10 (+0.78) | 62.64 (+2.65) | 13.36 (+1.01) |
| | OccFormer | Base | 37.48 | 48.71 | 61.92 | 12.83 |
| | | Our | 41.64 (+4.16) | 53.99 (+5.28) | 64.54 (+2.62) | 14.56 (+1.73) |
| | CGFormer | Base | 44.92 | 60.91 | 63.12 | 15.82 |
| | | Our | 48.64 (+3.72) | 67.35 (+6.44) | 63.53 (+0.41) | 17.02 (+1.20) |
| KITTI360 | VoxFormer | Base | 38.76 | 57.67 | 54.18 | 11.91 |
| | | Our | 43.25 (+4.49) | 65.81 (+7.29) | 55.78 (+1.60) | 13.55 (+1.64) |
| | CGFormer | Base | 48.21 | 69.98 | 60.78 | 19.11 |
| | | Our | 48.98 (+0.77) | 68.29 (-1.69) | 63.40 (+2.62) | 19.52 (+0.41) |

**Metric.** For OCC evaluation, we use intersection over union (IoU) to assess geometric completion, independent of semantics, as it is critical for obstacle avoidance in AVs. Semantic performance is measured by mean IoU (mIoU) across all classes. Given the strong negative correlation between IoU and mIoU (Li et al. (2023b)), an effective model must perform well on both metrics.

The experimental results of our Depth-UP on VoxFormer, OccFormer and CGFormer are presented in Table 1. As OccFormer is not available on KITTI360, we evaluate it with our Depth-UP only on SemanticKITTI. These results demonstrate that Depth-UP effectively leverages quantified uncertainty from the depth model to enhance OCC model performance, achieving up to a 4.49 (11.58%) improvement in IoU and up to a 1.73 (12.95%) improvement in mIoU, while also significantly improving both precision and recall in the geometry completion aspect of OCC. Even minor improvements in IoU and mIoU indicate meaningful progress in OCC performance (Zhang et al. (2023); Huang et al. (2023)). A detailed breakdown of mIoU results for each class is provided in Appendix A.7.

Figure 4 presents visualizations of the VoxFormer with and without our Depth-UP on SemanticKITTI. As highlighted by the orange dashed boxes, Depth-UP notably enhances the performance of OCC models in predicting rare classes, such as persons and bicyclists. Specifically, in the third row, Depth-UP successfully detects a person crossing the road at the corner, preventing a potential crash, while the baseline model fails

to recognize this individual. By improving the detection of such critical instances, our Depth-UP significantly reduces the risk of injury to pedestrians and enhances the safety of autonomous vehicles. Additional visualization results can be found in Appendix A.7.

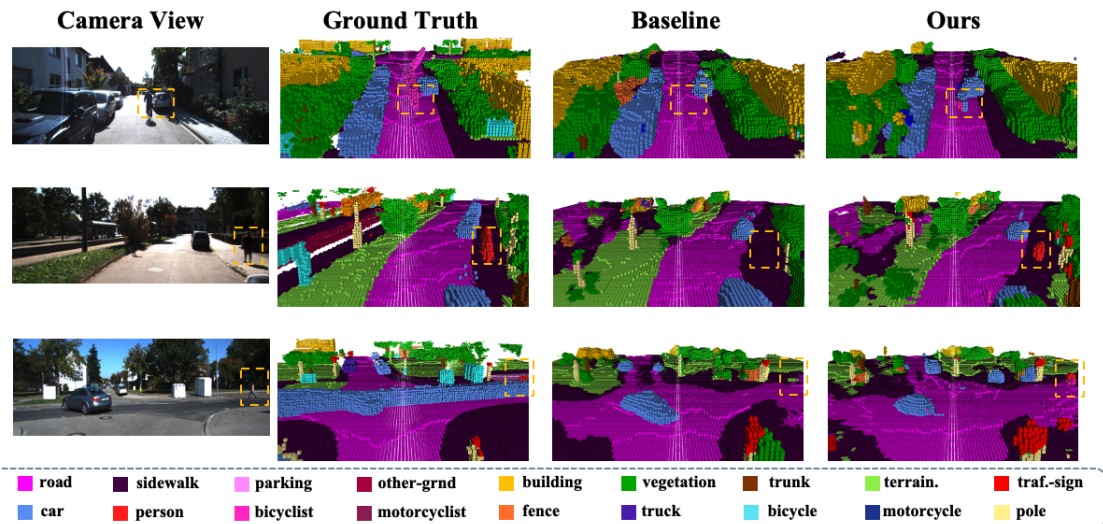

Figure 4: Qualitative results of the base VoxFormer model and that with our Depth-UP[1].

## 4.2 Uncertainty Quantification Performance

We evaluate our HCP on the geometric level and the final uncertainty quantification. Since we do not have the labeled test part of SemanticKITTI, we randomly split the original validation part of SemanticKITTI into the calibration dataset (take up 30%) and the test dataset (take up 70%). For KITTI360, we use the validation part as the calibration dataset and the test part as the test dataset.

**Geometric Level.** For the geometric level, the target of methods is to achieve the best trade-off between IoU and the occupied recall of rare classes. To show the effectiveness of our novel KL-based score function on the geometric level, we compare it with two common score functions in Angelopoulos & Bates (2021): class score $(1 - f(\mathbf{X})_y)$ and occupied score $(1 - \sum_{y=2}^{M} f(\mathbf{X})_y)$. Figure 5(a) shows the IoU results across different occupied recalls of the rare class person for different datasets. Figure 5(b) shows the IoU results across different occupied recalls of the rare class bicyclist for different basic OCC models. Here "Our Depth-UP" means the basic OCC model with our Depth-UP method. We can see that our KL-based score function always achieves the best geometry performance for the same occupied recall, compared with two baselines. Our HCP significantly outperforms baselines because it not only considers the occupied probability across all nonempty classes but also leverages the entire probability distribution.

To achieve the optimal balance between IoU and occupied recall, we can adjust the desired occupied recall. For instance, in the top right subfigure of Figure 5(a), the OCC without HCP shows an IoU of 45.85 and an occupied recall for the person class of 20.69. By setting the occupied recall to 21.75, the IoU improves to 45.94. Increasing the occupied recall beyond 30 (45.0% improvement) results in a decrease in IoU to 44.38 (3.4% reduction). This demonstrates that our HCP method can substantially boost the occupied recall of rare classes with a minor reduction in IoU.

**Uncertainty Quantification.** To measure the quantified uncertainty of different CP methods, we usually use the average class coverage gap (CovGap) and average set size (AvgSize) of the prediction sets as metrics (Ding et al. (2024)). For a given class $y \in \mathcal{Y} \setminus \{1\}$ with the defined error rate $\alpha^y$, the empirical class-conditional coverage of class $y$ is $c_y = \frac{1}{|\Upsilon^y|} \sum_{i \in \Upsilon^y} \mathbb{I}\{\mathbf{Y}_i \in \mathcal{C}(\mathbf{X}_i)\}$. The CovGap is defined as $\frac{1}{|\mathcal{Y}|-1} \sum_{y \in \mathcal{Y} \setminus \{1\}} |c_y - (1 - \alpha^y)|$. This measures how far the class-conditional coverage is from the desired coverage $1 - \alpha^y$. The AvgSize is

---

[1]The incorrect ground truth in the third row occurs because SemanticKITTI uses LiDAR temporal fusion for annotations, which results in ghosting effects for dynamic objects.

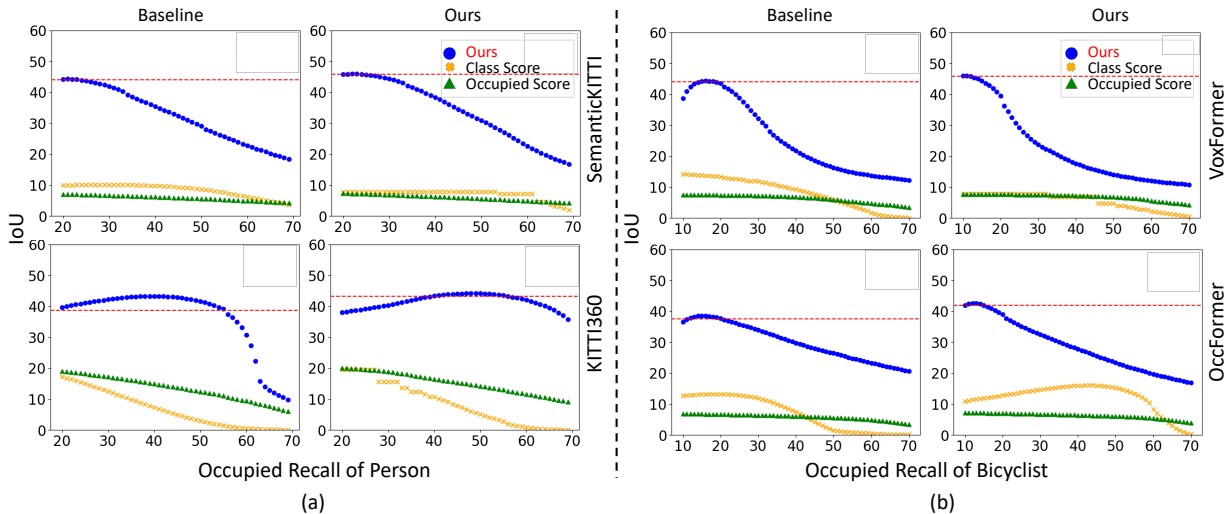

Figure 5: Compare our KL-based score function with the class and occupied scores. Evaluate OCC's geometry performance across different occupied recalls of the rare class. The red dotted line shows the IoU of the OCC without CP. (a): Results on the basic VoxFormer across different datasets for the person class. (b): Results on SemanticKITTI across different basic OCCs for the bicyclist class.

defined as $\frac{1}{T}\sum_{i=1}^{T}|\mathcal{C}(\mathbf{X}_i)|$, where $T$ is the number of samples in the test dataset and $\mathcal{C}(\mathbf{X}_i)$ does not contain the empty class. A good UQ method should achieve both small CovGap and AvgSize.

Table 2 compares our HCP method with SCP and CCCP on the SemanticKITTI dataset, as introduced in Subsection 3.2.1. For each class, the desired error rate is set by scaling the OCC model's original error rate by the scale $\lambda = 0.86$, raising the coverage requirement. The desired error rate of "Ours*" is the same as that on the base OCC model. The corresponding results on KITTI360 dataset can be found in Appendix A.8.

Our results demonstrate that HCP consistently achieves robust empirical class-conditional coverage and produces smaller prediction sets. In contrast, the performance of SCP and CCCP varies across different OCC models. Specifically, for our Depth-UP based on CGFormer, HCP reduces the set size by 90% and the coverage gap by 54%, compared to SCP. For our Depth-UP based on VoxFormer, HCP reduces the set size by 79% and the coverage gap by 64%, compared to CCCP. As noted in Subsection 3.2.1, SCP consistently fails to provide conditional coverage, although sometimes it provides a very small set size. Both SCP and CCCP tend to generate nonempty $C(\mathbf{X})$ for most voxels, potentially obstructing AVs. In contrast, HCP only generates nonempty $C(\mathbf{X})$ for these selected occupied voxels, thereby minimizing prediction set sizes while maintaining reliable class-conditional coverage. Moreover, under the same error rate setting, our HCP with the Depth-UP model (referred to as "Ours*") achieves the best UQ performance, reducing the set size by up to 18% under a similar coverage gap compared to applying HCP alone. This demonstrates that our $\alpha$-OCC framework, which integrates Depth-UP and HCP, achieves superior performance in both accuracy and UQ.

Table 2: For the SemanticKITTI, compare our HCP (referred to as "Ours") with SCP and CCCP on CovGap and AvgSize. "Ours*" are results of our HCP + Depth-UP under the same defined error rate for the base model.

| Basic OCC | | VoxFormer | | | | OccFormer | | | | CGFormer | | |
|---|---|---|---|---|---|---|---|---|---|---|---|---|
| Method | | Base | | Our Depth-UP | | Base | | Our Depth-UP | | Base | | Our Depth-UP |
| CP | SCP | CCCP | Ours | Ours* | SCP | CCCP | Ours | SCP | CCCP | Ours | Ours* | SCP | CCCP | Ours | SCP | CCCP | Ours* | SCP | CCCP | Ours |
| CovGap ↓ | 0.22 | 0.03 | 0.04 | 0.04 | 0.26 | 0.11 | 0.04 | 0.26 | 0.03 | 0.04 | 0.04 | 0.31 | 0.04 | 0.03 | 0.22 | 0.03 | 0.02 | 0.03 | 0.16 | 0.03 | 0.03 |
| AvgSize ↓ | 1.53 | 1.71 | 1.13 | 0.93 | 0.97 | 6.43 | 1.36 | 0.10 | 3.42 | 0.94 | 0.80 | 0.10 | 2.96 | 1.24 | 3.63 | 3.00 | 1.67 | 1.33 | 5.19 | 3.31 | 1.72 |

### 4.3 Ablation Study

Table 3: Ablation study on our Depth-UP framework with VoxFormer and SemanticKITTI.

| PGC | PSS | IoU ↑ | Precision ↑ | Recall ↑ | mIoU ↑ | FPS ↑ |
|---|---|---|---|---|---|---|
| | | 44.02 | 62.32 | 59.99 | 12.35 | **8.85** |
| ✓ | | 44.91 | 63.76 | 60.30 | 12.58 | 7.14 |
| | ✓ | 44.40 | 62.69 | 60.35 | 12.77 | 8.76 |
| ✓ | ✓ | **45.85** | 63.10 | **62.64** | **13.36** | 7.08 |

**Uncertainty Propagation.** We conducted an ablation study to assess the contributions of each technique proposed in our Depth-UP, as detailed in Table 3. The best results are shown in bold. The results indicate that Propagation on Geometry Completion (PGC) significantly enhances IoU, precision, and recall, which are key metrics for geometry. This improvement demonstrates how probabilistic geometry projection effectively utilizes uncertainty during the geometry stage of OCC. Additionally, Propagation on Semantic Segmentation (PSS) markedly improves mIoU, a crucial metric for semantic accuracy, demonstrating that uncertainty is effectively leveraged in semantic segmentation through depth feature extraction. Notably, the combined application of both techniques yields performance improvements that surpass the sum of their individual contributions. This shows that uncertainty propagation in OCC completion and segmentation boosts performance. Additional results on the influence of depth mean can be found in Appendix A.7.

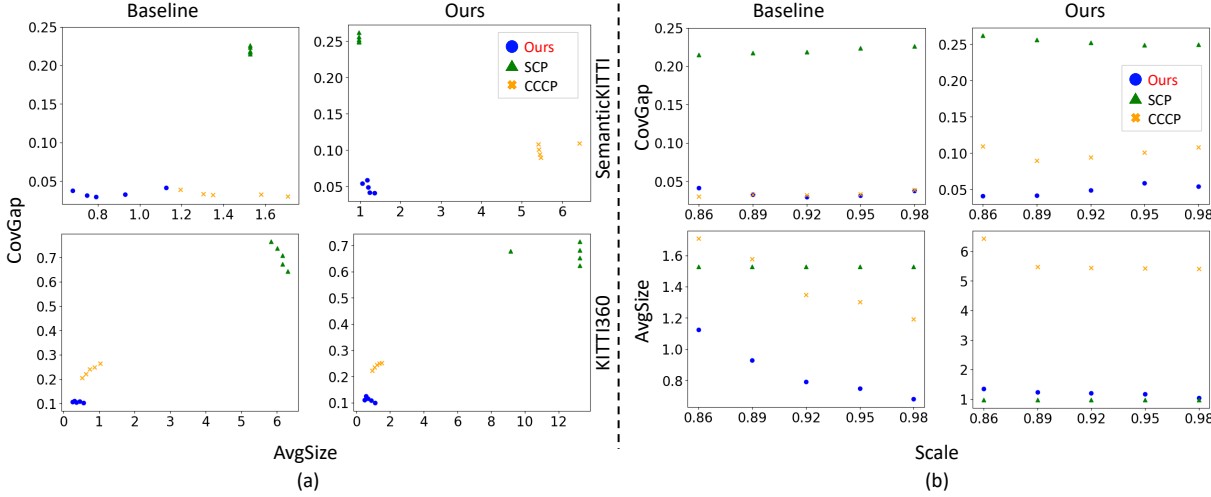

Figure 6: Compare our HCP with SCP and CCCP on CovGap and AvgSize with VoxFormer. Each point denotes a desired class error rate. Lower CovGap and AvgSize indicate better performance. (a): CovGap vs. AvgSize on different settings across different datasets. (b): CovGap vs. scale and AvgSize vs. scale on the SemanticKITTI where the scale represents the desired class error rate.

**Uncertainty Quantification.** We compare our HCP with SCP and CCCP under different desired class-specific error rate $\alpha^y$ settings with the basic VoxFormer, as shown in Figure 6. We consider five error rate scale settings with $\lambda \in \{0.86, 0.89, 0.92, 0.95, 0.98\}$. The points of our HCP are always located in the left bottom corner of subfigures in Figure 6(a) which means our HCP achieves the best performance on set size and coverage gap under all error rate settings. In Figure 6(b), our HCP always achieves low CovGap indicating it can always satisfy the coverage guarantee even under high requirements. SCP consistently has the largest CovGap due to its disregard for conditional coverage. While CCCP addresses this issue, it often results in a high AvgSize because it generates prediction sets for empty voxels. In contrast, our HCP method not only accounts for conditional coverage but also excludes empty voxels, generating prediction sets exclusively for predicted occupied voxels. For all CP approaches, as the desired error rate becomes smaller, the set size tends to be larger. CPs increase the set size to satisfy the coverage guarantee. The results on

other OCC models are shown in Appendix A.8, where our HCP is applied to one LiDAR-based OCC to show its scalability.

**Limitation.** Our Depth-UP introduces a 20% reduction in FPS; however, this does not substantially impact the overall efficiency of OCC models. Notably, we did not apply any code-level optimization to improve runtime, indicating the added computational overhead is acceptable.

# 5 Conclusion

This paper introduces a novel approach to enhancing camera-based 3D Semantic Occupancy Prediction by incorporating uncertainty inherent in models. Our proposed framework, $\alpha$-OCC, integrates the Depth-UP from depth models to improve OCC performance in both geometry completion and semantic segmentation. A novel HCP method is designed to quantify OCC uncertainty effectively under high-level class imbalance. Our extensive experiments demonstrate the effectiveness of $\alpha$-OCC. The Depth-UP significantly improves prediction accuracy, achieving up to 11.58% increase in IoU and up to 12.95% increase in mIoU. The HCP significantly improves performance by ensuring robust class-conditional coverage while keeping compact prediction sets. Compared to baselines, it achieves up to 90% reduction in set size and decreases the coverage gap by up to 64%. Additionally, integrating HCP with Depth-UP further reduces the set size by 18%. These results highlight the significant improvements in both accuracy and uncertainty quantification offered by our approach, especially for rare safety-critical classes, such as persons and bicyclists, thereby reducing potential risks for AVs. In the future, we will extend HCP to other highly imbalanced classification tasks.

### Acknowledgments

Sanbao Su, Chenchen Lin and Fei Miao are supported by the National Science Foundation under Grants CNS-2047354 (CAREER). Chen Feng is supported by the National Science Foundation under Grants 2514030 and 2238968 (CAREER).

This material is based upon work supported under the AI Research Institutes program by National Science Foundation and the Institute of Education Sciences, U.S. Department of Education through Award # 2229873 - National AI Institute for Exceptional Education. Any opinions, findings and conclusions or recommendations expressed in this material are those of the author(s) and do not necessarily reflect the views of the National Science Foundation, the Institute of Education Sciences, or the U.S. Department of Education.

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

# A    Appendix

## A.1    More Related Work

**Class Imbalance.** In real-world applications like robotics and autonomous vehicles (AVs), datasets often face the challenge of class imbalance (Chen et al. (2018)). Rare classes, typically encompassing high safety-critical entities such as persons, are significantly outnumbered by lower safety-critical classes like trees and buildings. Various strategies have been proposed to tackle class imbalance. Data-level methods involve random under-sampling of majority classes and over-sampling of minority classes during training (Van Hulse et al. (2007)). However, they struggle to address the pronounced class imbalance encountered in OCC (Megahed et al. (2021)), as shown in Section 1. Algorithm-level methods employ cost-sensitive losses to adjust the training process for different tasks, such as depth estimation (Eigen & Fergus (2015)) and 2D segmentation (Badrinarayanan et al. (2017)). While algorithm-level methods have been widely implemented in current OCC models (Voxformer Li et al. (2023b) utilizes Focal Loss Lin et al. (2017) as the loss function), they still fall short in accurately predicting minority classes. In contrast, classifier-level methods postprocess output class probabilities during the testing phase through posterior calibration (Buda et al. (2018); Tian et al. (2020)). In this paper, we propose a hierarchical conformal prediction method falling within this category, aimed at enhancing the recall of rare safety-critical classes in the OCC task.

## A.2    Introduction on Semantic Occupancy Prediction Models

**Semantic Occupancy Prediction.** The concept of 3D Semantic Occupancy Prediction (OCC), which is also known as 3D semantic scene completion, was first introduced by SSCNet (Song et al. (2017)), integrating both geometric and semantic reasoning. Since its inception, numerous studies have emerged, categorized into two streams: LiDAR-based OCC (Roldao et al. (2020); Cheng et al. (2021)) and camera-based OCC (Cao & De Charette (2022); Li et al. (2023b); Zhang et al. (2023); Huang et al. (2024); Vobecky et al. (2024)).

Recently, camera-based OCC has gained increasing attention due to its advantages in visual recognition and cost-effectiveness (Ma et al. (2024)). Depth predictions from depth models are instrumental in projecting 2D information into 3D space for it. Existing methodologies can be classified into two paradigms based on their utilization of depth information: 3D-to-2D querying and 2D-to-3D lifting. The former Li et al. (2023b; 2022) generates query proposals using depth estimation and leverages them to extract rich visual features from the 3D scene. The latter Tian et al. (2024); Zhang et al. (2023); Yu et al. (2024), meanwhile, projects multi-view 2D image features into depth-aware frustums, as proposed by LSS Philion & Fidler (2020).

However, these methods overlook depth estimation uncertainty. Despite leveraging latent depth distribution, 2D-to-3D lifting technique sacrifices precise information and neglects lens distortion issues during geometry completion (Lucas et al. (2013)). To address this, we propose an uncertainty propagation framework from depth models to enhance OCC performance.

During the experiments, we used three OCC models: VoxFormer (Li et al. (2023b)), OccFormer (Zhang et al. (2023)) and CGFormer (Yu et al. (2024)). VoxFormer is the 3D-to-2D querying approach. OccFormer and CGFormer are the 2D-to-3D lifting approaches. So we have considered both paradigms that utilize depth information on OCC models in our experiments.

Recent advancements have explored leveraging diverse data sources and training strategies to enhance occupancy accuracy. 2DPASS (Yan et al. (2022)) demonstrates the utility of using 2D priors to assist 3D semantic segmentation on LiDAR point clouds. EPMF (Tan et al. (2024)) focuses on efficient multi-sensor fusion for robust perception. And Recent works like Co-Occ (Pan et al. (2024a)) and RadOcc (Zhang et al. (2024)) introduce volume rendering and knowledge distillation to couple explicit feature fusion with global scene consistency in OCC. While these methods improve base performance, they remain primarily deterministic. Our OCC framework is complementary to these architectures, providing a layer of uncertainty awareness that enhances robustness for OCC.

### A.3 Algorithm of HCP

Algo. 1 shows the detailed algorithm of our HCP.

### A.4 Proof of Proposition 1

**Proposition 1.** For a desired $\alpha^y$ value, we select $\alpha_o^y$ and $\alpha_s^y$ as $1 - \alpha^y = (1 - \alpha_s^y)(1 - \alpha_o^y)$, then the prediction set generated as Eq. 4 satisfies that $\mathbb{P}(\mathbf{Y}_{test} \in \mathcal{C}(\mathbf{X}_{test})|\mathbf{Y}_{test} = y) \geq 1 - \alpha^y$.

*Proof.*

$$\mathbb{P}(\mathbf{Y}_{test} \in \mathcal{C}(\mathbf{X}_{test})|\mathbf{Y}_{test} = y) = \sum_o \mathbb{P}(\mathbf{Y} \in \mathcal{C}(\mathbf{X})_{test})|\mathbf{Y}_{test} = y, o)\mathbb{P}(o|\mathbf{Y}_{test} = y)$$

$$=\mathbb{P}(\mathbf{Y}_{test} \in \mathcal{C}(\mathbf{X}_{test})|\mathbf{Y}_{test} = y, o = T)\mathbb{P}(o = T|\mathbf{Y}_{test} = y)$$
$$+ \mathbb{P}(\mathbf{Y}_{test} \in \mathcal{C}(\mathbf{X}_{test})|\mathbf{Y}_{test} = y, o = F)\mathbb{P}(o = F|\mathbf{Y}_{test} = y)$$
$$=\mathbb{P}(\mathbf{Y}_{test} \in \mathcal{C}(\mathbf{X}_{test})|\mathbf{Y}_{test} = y, o = T)\mathbb{P}(o = T|\mathbf{Y}_{test} = y) \geq (1 - \alpha_s^y)(1 - \alpha_o^y)$$
$$\Rightarrow\mathbb{P}(\mathbf{Y}_{test} \in \mathcal{C}(\mathbf{X}_{test})|\mathbf{Y}_{test} = y) \geq 1 - \alpha^y, \texttt{when } 1 - \alpha^y = (1 - \alpha_s^y)(1 - \alpha_o^y)$$

$\square$

### A.5 Introduction on Datasets

During the experiments, we used two datasets: SemanticKITTI (Behley et al. (2019)) and KITTI360 (Li et al. (2023a)). SemanticKITTI provides dense semantic annotations for each LiDAR sweep composed of 22 outdoor driving scenarios based on the KITTI Odometry Benchmark (Geiger et al. (2012)). Regarding the sparse input to an OCC model, it can be either a single voxelized LiDAR sweep or an RGB image. The voxel grids are labeled with 20 classes (19 semantics and 1 empty), with the size of 0.2m × 0.2m × 0.2m. We only used the train and validation parts of SemanticKITTI as the annotations of the test part are not available. SSCBench-KITTI-360 provides dense semantic annotations for each image based on KITTI360 (Liao et al. (2022)), which is also called KITTI360 for simplification. The voxel grids are labeled with 19 classes (18 semantics and 1 empty), with the size of 0.2m × 0.2m × 0.2m. Both SemanticKITTI and KITTI360 are interested in a volume of 51.2m ahead of the car, 25.6m to left and right side, and 6.4m in height.

### A.6 Experimental Setting

We used two different servers to conduct experiments on the SemanticKITTI and KITTI360 datasets. For the SemanticKITTI dataset, we employed a system equipped with four NVIDIA Quadro RTX 8000 GPUs, each providing 48GB of VRAM. The system was configured with 128GB of system RAM. The training process required approximately 30 minutes per epoch, culminating in a total training duration of around 16 hours for 30 epochs. The software environment included the Linux operating system (version 18.04), Python 3.8.19, CUDA 11.1, PyTorch 1.9.1+cu111, and CuDNN 8.0.5.

For the KITTI360 dataset, we used a different system equipped with eight NVIDIA GeForce RTX 4090 GPUs, each providing 24GB of VRAM, with 720GB of system RAM. The training process required approximately 15 minutes per epoch, culminating in a total training duration of around 8 hours for 30 epochs. The software environment comprised the Linux operating system(version 18.04), Python 3.8.16, CUDA 11.1, PyTorch 1.9.1+cu111, and CuDNN 8.0.5. These settings ensure the reproducibility of our experiments on similar hardware configurations.

In our training, we used the AdamW optimizer with a learning rate of 2e-4 and a weight decay of 0.01. The learning rate schedule followed a Cosine Annealing policy with a linear warmup for the first 500 iterations, starting at a warmup ratio of $\frac{1}{3}$. The minimum learning rate ratio was set to 1e-3. We applied gradient clipping with a maximum norm of 35 to stabilize the training.

---

**Algorithm 1:** Our HCP

---

**Data:** number of classes is $M$, calibration dataset $\mathcal{D}_{cali}(\mathbf{X}, \mathbf{Y})$ with $N$ samples, test dataset $\mathcal{D}_{test}(\mathbf{X})$
  with $T$ samples, the considered rare class set $\mathcal{Y}_r$, the occupied error rate $\alpha_o^y \ \forall y \in \mathcal{Y}_r$ ,desired
  class-specific error rate $\alpha^y \ \forall y \in \mathcal{Y} \backslash \{1\}$, the OCC model $f$.

**Result:** Prediction set $\mathcal{C}(\mathbf{X}_i)$, $\forall \mathbf{X}_i \in \mathcal{D}_{test}$

1 /* Calibration Step:  Geometric Level                                                 */

2 $\mathcal{S}^y = \emptyset \ \forall y \in \mathcal{Y}_r$; $\mathbf{O} = \{\varepsilon, 1, ..., 1\}^M$;

3 **for** $(\mathbf{X}_i, \mathbf{Y}_i) \in \mathcal{D}_{cali}$ **do**

4      $s_{kl}(\mathbf{X}, y) = \mathrm{D}_{kl}(f(\mathbf{X}_i) || \mathbf{O}) \ y = \mathbf{Y}_i \in \mathcal{Y}_r$ as Eq. 2; add $s_{kl}(\mathbf{X}, y)$ into $\mathcal{S}^y$;

5 **end**

6 $q_o^y = \mathrm{Quantile}(\frac{\lceil (N_y+1)(1-\alpha_o^y) \rceil}{N_y}, \mathcal{S}^y)$ where $N_y = |\mathcal{S}^y|$, $\forall y \in \mathcal{Y}_r$;

7 /* Calibration Step:  Semantic Level                                                */

8 $\mathcal{S}_o^y = \emptyset$, $tp_y = 0$ and $fn_y = 0 \ \forall y \in \mathcal{Y} \backslash \{1\}$;

9 **for** $(\mathbf{X}_i, \mathbf{Y}_i) \in \mathcal{D}_{cali}$ *and* $\mathbf{Y}_i \in \mathcal{Y} \backslash \{1\}$ **do**

10      **if** $\exists y \in \mathcal{Y}_r$, $s_{kl}(\mathbf{X}_i, y) \le q_o^y$ **then**

11          add $1 - f(\mathbf{X}_i)_{\mathbf{Y}_i}$ into $\mathcal{S}_o^{\mathbf{Y}_i}$ and $tp_{\mathbf{Y}_i} = tp_{\mathbf{Y}_i} + 1$;

12      **else**

13          $fn_{\mathbf{Y}_i} = fn_{\mathbf{Y}_i} + 1$;

14      **end**

15 **end**

16 **for** $y \in \mathcal{Y} \backslash \{1\}$ **do**

17      $\alpha_o^y = 1 - \frac{tp_y}{tp_y + fn_y}$ if $y \notin \mathcal{Y}_r$

18      $\alpha_s^y = 1 - \frac{1-\alpha^y}{1-\alpha_o^y}$; $q_s^y = \mathrm{Quantile}(\frac{\lceil (N_{yo}+1)(1-\alpha_s^y) \rceil}{N_{yo}}, \mathcal{S}_o^y)$ where $N_o^y = |\mathcal{S}_o^y|$

19 **end**

20 /* Test Step                                                       */

21 **for** $\mathbf{X}_i \in \mathcal{D}_{test}$ **do**

22      **if** $\exists y \in \mathcal{Y}_r$, $s_{kl}(\mathbf{X}, y) \le q_o^y$ **then**

23          $\mathcal{C}(\mathbf{X}_i) = \{y : 1 - f(\mathbf{X}_i)_y \le q_s^y\}$

24      **else**

25          $\mathcal{C}(\mathbf{X}_i) = \emptyset$ which means it is empty class.

26      **end**

27 **end**

---

The user-defined target error rate $\alpha^y$ for each class $y$ is decided according to the prediction error rate of the original model. For each class, It is set by multiplying the original prediction error rate of OCC models with the scale $\lambda < 1$, which raises the coverage requirement. For example, for the person class, if the original model has 90% prediction error rate and we set the scale $\lambda = 0.9$, the user-defined target error rate $\alpha^{person}$ of person is decided as $90\% * 0.9 = 81\%$.

### A.7 More Results on Depth-UP

Table 5 presents a comparative analysis of our Depth-UP models against various OCC models, providing detailed mIoU results for different classes. Our Depth-UP demonstrates superior performance in geometry completion and semantic segmentation, outperforming all other OCC models and even surpassing LiDAR-based OCC models on the SemanticKITTI dataset. The VoxFormer with our Depth-UP achieves the best IoU on SemanticKITTI and the OccFormer with our Depth-UP achieves the best mIoU on SemanticKITTI. This improvement is attributed to propagating the inherent uncertainty of the depth model into both geometry completion and semantic segmentation of OCCs. Notably, on the KITTI360 dataset, our Depth-UP achieves the highest mIoU for bicycle, motorcycle, and person classes, which are crucial for safety.

For the person and bicyclist categories on the SemanticKITTI dataset, our Depth-UP decreases the mIoU. This issue primarily stems from annotation defects, particularly for dynamic objects such as persons and bicyclists. The SemanticKITTI dataset generates annotations using LiDAR temporal fusion, which introduces ghosting effects for moving objects. This problem has been documented in Figure 2 of the SSCBench (Li et al. (2023a)). While cars are also affected, most are stationary, so the impact is minimal. However, nearly all persons and bicyclists in the SemanticKITTI validation set are moving, leading to erroneous annotations. SSCBench has acknowledged this issue, and thus KITTI360 proposed in SSCBench does not suffer from ghosting problems. Our Depth-UP shows mIoU improvements in both person and bicyclist categories on KITTI360, aligning with our expectations. This may also explain why our Depth-UP enhances VoxFormer significantly more on KITTI360 compared to SemanticKITTI, showing a 1.64 mIoU improvement versus a 1.01 mIoU improvement.

**Visualization.** Figure 7 provides additional visualizations of the OCC model's performance with and without our Depth-UP on the SemanticKITTI dataset. These visualizations demonstrate that our Depth-UP enhances the model's ability to predict rare classes, such as persons and bicyclists, which are highlighted with orange dashed boxes. Notably, in the fourth row, our Depth-UP successfully predicts the presence of a person far from the camera, whereas the baseline model fails to do so. This indicates that Depth-UP improves object prediction in distant regions. By enhancing the detection of such critical objects, our Depth-UP significantly reduces the risk of accidents, thereby improving the safety of autonomous vehicles.

**Ablation Study.** We further conducted the ablation study on the influence of depth information on the semantic feature part of our Depth-UP framework, as shown in Table 4. When Propagation on Geometry Completion (PGC) is not applied, using only the depth mean in the Propagation on Semantic Segmentation (PSS, second row) improves IoU by 0.85 and mIoU by 0.16. When PGC is applied, compared to using only PGC (fourth row), incorporating only the depth mean in PSS (fifth row) increases IoU by 0.45 and mIoU by 0.08. These results indicate that the depth mean enhances geometric performance (IoU) and slightly improves semantic performance (mIoU). In contrast, incorporating depth uncertainty into PSS further improves mIoU by 0.26 when PGC is not applied and by 0.7 when PGC is applied. This demonstrates that leveraging depth uncertainty in PSS significantly enhances semantic performance. The depth mean primarily accounts for the improvement in IoU, suggesting a partial decoupling of the effects of uncertainty. One additional observation is that when PGC is not applied, PSS with only the depth mean achieves a higher IoU than the complete PSS, due to an imbalance in precision and recall. However, since the complete Depth-UP achieves the best performance in both IoU and mIoU across all settings, incorporating depth uncertainty into semantic features remains essential for optimal performance.

**Discussion.** The depth mean estimation $\hat{\mathbf{D}}$ in our OCC model remains identical to that in the corresponding base OCC model, as we only retrained the additional regression head for $\hat{\boldsymbol{\Sigma}}$ in the depth model. The improvement achieved by our OCC model does not result from nonexistent enhancements to the depth mean estimation; rather, it arises from effectively incorporating the uncertainty of depth estimation into the OCC models.

Table 4: Ablation study on the depth information for our Depth-UP framework with VoxFormer and SemanticKITTI.

| PGC | PSS | IoU ↑ | Precision ↑ | Recall ↑ | mIoU ↑ | FPS ↑ |
|---|---|---|---|---|---|---|
| | | 44.02 | 62.32 | 59.99 | 12.35 | **8.85** |
| | mean | 44.87 | 59.8 | 64.25 | 12.51 | 8.77 |
| | mean & std | 44.40 | 62.69 | 60.35 | 12.77 | 8.76 |
| ✓ | | 44.91 | 63.76 | 60.30 | 12.58 | 7.14 |
| ✓ | mean | 45.36 | 62.76 | 62.06 | 12.66 | 7.08 |
| ✓ | mean & std | **45.85** | **63.10** | **62.64** | **13.36** | 7.08 |

Table 5: **Separate results on SemanticKITTI and KITTI360.** We evaluate our Depth-UP models on two datasets. The default evaluation range is $51.2{\times}51.2{\times}6.4\text{m}^3$. Due to the label differences between the two subsets, missing labels are replaced with "-". "Depth-UP*" means the VoxFormer with our Depth-UP method. "Depth-UP†" means the OccFormer with our Depth-UP method. "Depth-UP‡" means the CGFormer with our Depth-UP method. The top three performances on each dataset are marked by red, green, and blue respectively.

| Dataset | Method | Input | IoU | mIoU | car | bicycle | motorcycle | truck | other-veh. | person | road | parking | sidewalk | other-grnd | building | fence | vegetation | terrain | pole | traf.-sign | bicyclist | trunk | motorcyclist |
|---|---|---|---|---|---|---|---|---|---|---|---|---|---|---|---|---|---|---|---|---|---|---|---|
| SemanticKITTI | LMSCNet | L | 46.74 | 14.77 | 30.58 | 0.00 | 0.00 | 2.11 | 0.08 | 0.00 | 57.41 | 12.86 | 31.86 | 0.00 | 34.69 | 9.32 | 31.38 | 38.86 | 19.52 | 0.43 | 0.00 | 11.52 | 0.00 |
| | SSCNet | L | 40.93 | 10.27 | - | - | - | - | - | - | - | - | - | - | - | - | - | - | - | - | - | - | - |
| | MonoScene | C | 36.80 | 11.30 | 23.29 | 0.28 | 0.59 | 9.29 | 2.63 | 2.00 | 55.89 | 14.75 | 26.50 | 1.63 | 13.55 | 6.60 | 17.98 | 29.84 | 3.91 | 2.43 | 1.07 | 2.44 | 0.00 |
| | VoxFormer | C | 44.02 | 12.35 | 25.79 | 0.59 | 0.51 | 5.63 | 3.77 | 1.78 | 54.76 | 15.50 | 26.35 | 0.70 | 17.65 | 7.64 | 24.39 | 29.96 | 7.11 | 4.18 | 3.32 | 5.08 | 0.00 |
| | TPVFormer | C | 35.61 | 11.36 | 23.81 | 0.36 | 0.05 | 8.08 | 4.35 | 0.51 | 56.50 | 20.60 | 25.87 | 0.85 | 13.88 | 5.94 | 16.92 | 30.38 | 3.14 | 1.52 | 0.89 | 2.26 | 0.00 |
| | OccFormer | C | 36.50 | 13.46 | 25.09 | 0.81 | 1.19 | 25.53 | 8.52 | 2.78 | 58.85 | 19.61 | 26.88 | 0.31 | 14.40 | 5.61 | 19.63 | 32.62 | 4.26 | 2.86 | 2.82 | 3.93 | 0.00 |
| | CGFormer | C | 44.92 | 15.82 | 34.07 | 2.87 | 4.26 | 5.80 | 6.63 | 1.51 | 66.46 | 21.27 | 33.06 | 0.00 | 21.77 | 9.18 | 25.87 | 40.18 | 10.73 | 7.02 | 3.13 | 8.62 | 0.00 |
| | Depth-UP* (ours) | C | 45.85 | 13.36 | 28.51 | 0.12 | 3.57 | 12.01 | 4.23 | 2.24 | 55.72 | 14.38 | 26.20 | 0.10 | 20.58 | 7.70 | 26.24 | 30.26 | 8.03 | 5.81 | 1.18 | 7.03 | 0.00 |
| | Depth-UP† (ours) | C | 41.97 | 14.56 | 26.53 | 1.12 | 1.54 | 10.64 | 9.37 | 2.63 | 62.38 | 21.58 | 29.79 | 1.97 | 18.85 | 7.69 | 24.68 | 34.09 | 7.86 | 5.82 | 1.61 | 7.40 | 0.00 |
| | Depth-UP‡ (ours) | C | 48.64 | 17.02 | 34.84 | 2.43 | 6.40 | 5.26 | 10.07 | 2.26 | 66.47 | 20.47 | 33.12 | 0.26 | 27.36 | 10.11 | 29.82 | 42.72 | 12.12 | 7.71 | 2.86 | 8.52 | 0.00 |
| KITTI-360 | LMSCNet | L | 47.53 | 13.65 | 20.91 | 0 | 0 | 0.26 | 0 | 0 | 62.95 | 13.51 | 33.51 | 0.2 | 43.67 | 0.33 | 40.01 | 26.80 | 0 | 0 | - | - | - |
| | SSCNet | L | 53.58 | 16.95 | 31.95 | 0 | 0.17 | 10.29 | 0.58 | 0.07 | 65.7 | 17.33 | 41.24 | 3.22 | 44.41 | 6.77 | 43.72 | 28.87 | 0.78 | 0.75 | - | - | - |
| | MonoScene | C | 37.87 | 12.31 | 19.34 | 0.43 | 0.58 | 8.02 | 2.03 | 0.86 | 48.35 | 11.38 | 28.13 | 3.22 | 32.89 | 3.53 | 26.15 | 16.75 | 6.92 | 5.67 | - | - | - |
| | VoxFormer | C | 38.76 | 11.91 | 17.84 | 1.16 | 0.89 | 4.56 | 2.06 | 1.63 | 47.01 | 9.67 | 27.21 | 2.89 | 31.18 | 4.97 | 28.99 | 14.69 | 6.51 | 6.92 | - | - | - |
| | CGFormer | C | 48.21 | 19.11 | 29.82 | 3.77 | 4.95 | 12.77 | 8.39 | 6.74 | 63.52 | 18.52 | 40.71 | 4.71 | 42.84 | 8.34 | 38.82 | 24.30 | 16.51 | 19.29 | - | - | - |
| | Depth-UP* (ours) | C | 43.25 | 13.55 | 22.32 | 1.96 | 1.58 | 9.43 | 2.27 | 3.13 | 53.50 | 11.86 | 31.63 | 3.20 | 34.49 | 6.11 | 32.01 | 18.78 | 11.46 | 13.65 | - | - | - |
| | Depth-UP‡ (ours) | C | 48.98 | 19.53 | 29.63 | 4.10 | 4.07 | 16.80 | 7.88 | 6.05 | 63.12 | 18.12 | 40.70 | 5.80 | 43.74 | 9.02 | 39.47 | 24.53 | 17.99 | 20.49 | - | - | - |

Table 6: For the KITTI360 and VoxFormer, compare our HCP (referred to as "Ours") with SCP and CCCP on CovGap and AvgSize. "Ours*" are results of our HCP + Depth-UP under the same defined error rate for the base model.

| Method | Base | | | | Our Depth-UP | | |
|---|---|---|---|---|---|---|---|
| CP | SCP | CCCP | Ours | Ours* | SCP | CCCP | Ours |
| CovGap ↓ | 0.64 | 0.26 | 0.10 | 0.08 | 0.62 | 0.25 | 0.10 |
| AvgSize ↓ | 6.30 | 1.03 | 0.56 | 0.49 | 13.24 | 1.51 | 1.12 |

## A.8 More Results on HCP

**Geometric Level.** Similar to Figure 5, we evaluate the geometry performance of our HCP module by comparing the proposed KL-based score function against standard class score and occupied score baselines for the bicycle and motorcycle classes. As shown in Figure 8, we assess the geometry performance (IoU) of the VoxFormer baseline across varying levels of occupied recall for these rare classes. The results demonstrate that our KL-based score function consistently maintains superior geometry performance across the entire range of occupied recalls for all safety-critical rare classes. By effectively capturing the nuances of the probability distribution through our hierarchical approach, the HCP ensures that these rare entities are recovered with significantly higher precision than traditional scores. This consistent performance across diverse classes and datasets underscores the robustness and generalizability of our uncertainty-aware framework in addressing the extreme class imbalance inherent in 3D occupancy tasks.

**More Discussion.** Our HCP significantly outperforms baselines because it not only considers the occupied probability across all nonempty classes but also leverages the entire probability distribution. Compared with the class score, which only considers individual class probabilities, our score function accounts for all nonempty classes. Predicting rare classes is challenging for models, but they tend to identify these as occupied, assigning lower probabilities to the empty class and higher probabilities to all nonempty classes. Therefore, it's crucial to consider the probability of all nonempty classes. Although the occupied score addresses this by summing probabilities of all nonempty classes, it loses sensitivity to the distribution. When facing difficult

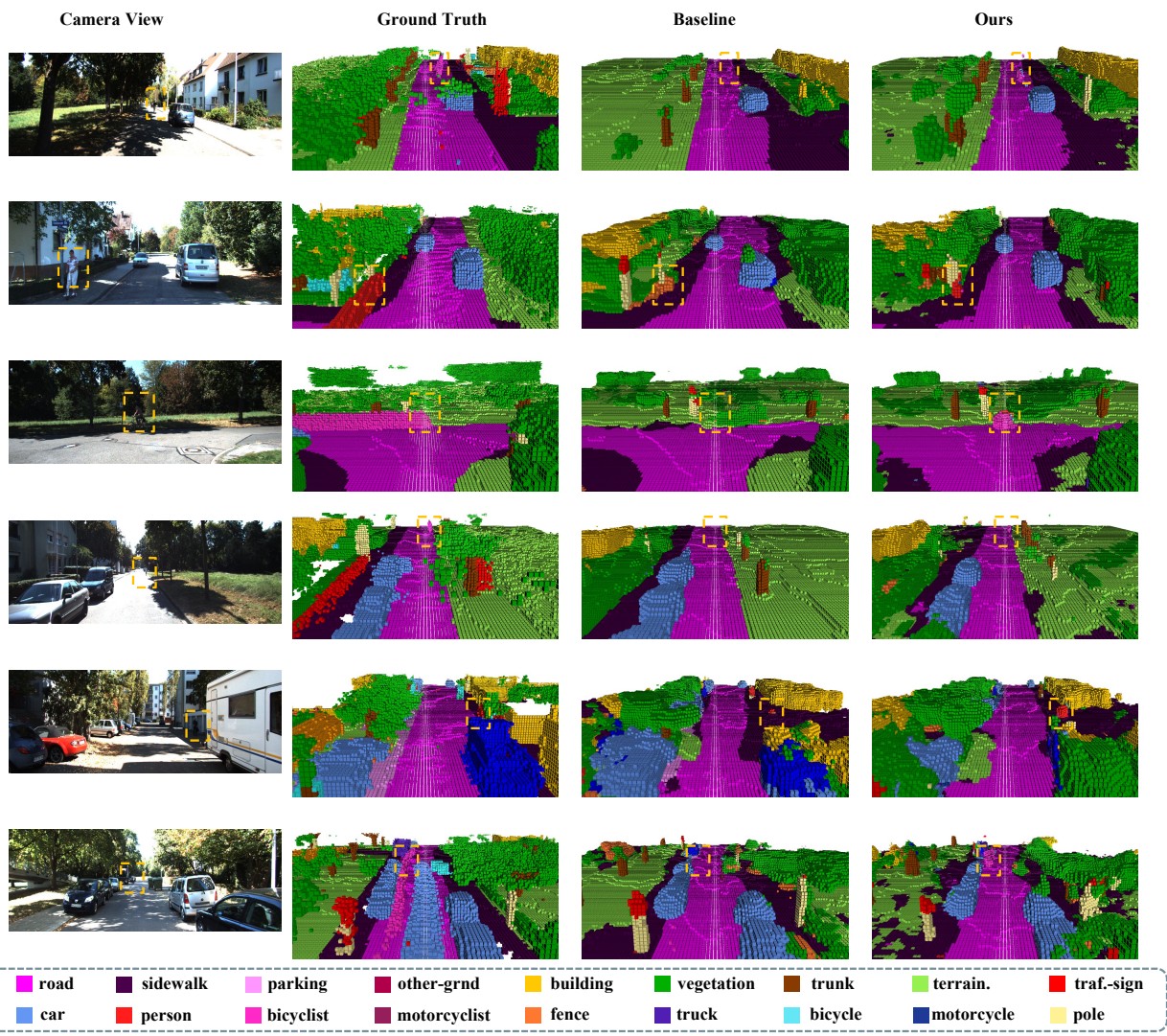

Figure 7: Qualitative results of the baseline OCC model and that with our Depth-UP method.

samples (such as rare classes), models tend to produce output probabilities that are more evenly distributed across the possible classes (Guo et al. (2017)). The Kullback-Leibler (KL) divergence measures how one probability distribution diverges from a reference distribution, considering the entire shape of the probability distribution (Raiber & Kurland (2017)). This sensitivity to distribution shape enables our KL-based score function to identify rare classes more effectively.

**Uncertainty Quantification.** Similar to Table 2, we compare our HCP method with SCP and CCCP on the KITTI-360 dataset in Table 6. For each class, the desired error rate is obtained by scaling the original error rate of the OCC model using a factor of $\lambda = 0.86$, thereby increasing the coverage requirement. The desired error rate for "Ours*" remains identical to that of the base OCC model. Our results demonstrate that HCP consistently achieves robust empirical class-conditional coverage while producing significantly smaller prediction sets. In contrast, the performance of SCP and CCCP varies across different OCC models. Specifically, for our Depth-UP model based on VoxFormer, HCP reduces the prediction set size by 92% and the coverage gap by 84% compared to SCP. For the base VoxFormer model, HCP reduces the set size by 62% and the coverage gap by 46% compared to CCCP.

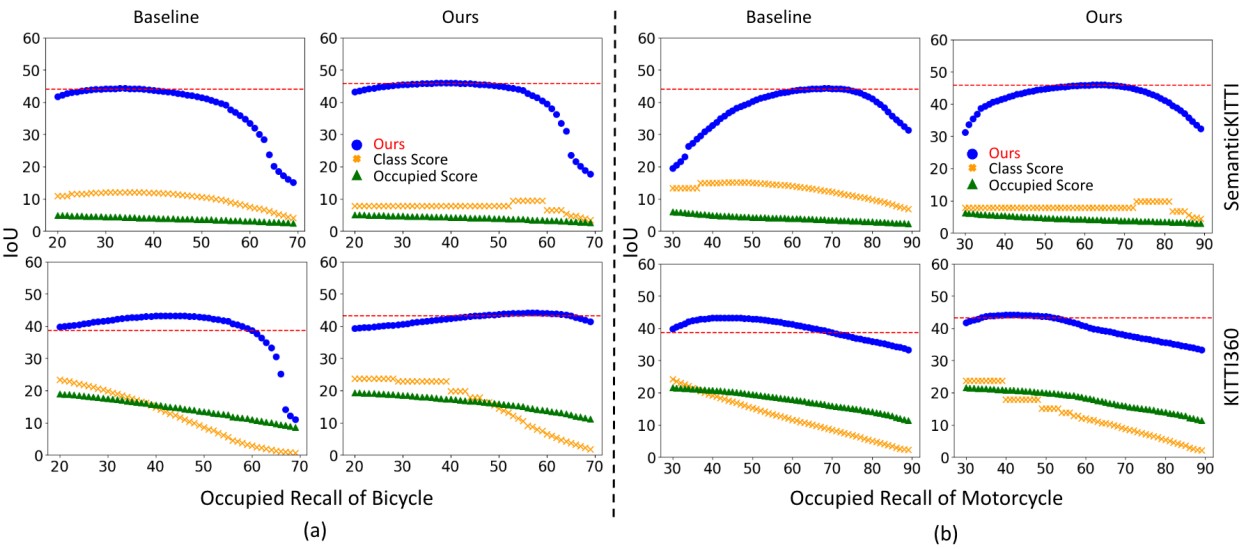

Figure 8: Compare our KL-based score function with the class and occupied scores. Evaluate OCC's geometry performance across different occupied recalls of the rare class on the basic VoxFormer across different datasets. The red dotted line shows the IoU of the OCC without CP. (a): Results for the bicycle class. (b): Results the motorcycle class.

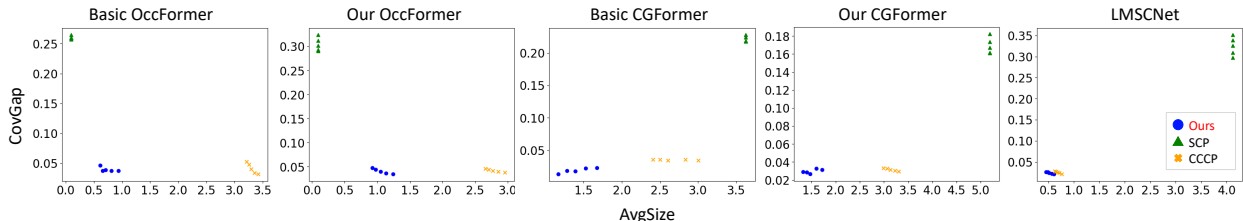

Figure 9: The results of CovGap vs, AvgSize for our HCP, SCP and CCCP on SemanticKITTI. The considered OCC models are the basic OccFormer, the OccFormer with our Depth-UP, the basic CGFormer, the CGFormer with our Depth-UP, and the LiDAR-based OCC model LMSCNet.

**Ablation Study.** We compare our HCP with SCP and CCCP under different desired class-specific error rate settings on more OCC models: the basic OccFormer, the OccFormer with our Depth-UP, the basic CGFormer, the CGFormer with our Depth-UP, and the LiDAR-based OCC model LMSCNet (Roldao et al. (2020)) to show the scalability of our HCP. The dataset used here is SemanticKITTI. For each class, the desired error rate is set by multiplying the original error rate of OCC models with the scale $\lambda$, $\lambda \in \{0.86, 0.89, 0.92, 0.95, 0.98\}$, which raises the coverage requirement. Figure 9 shows the CovGap vs, AvgSize results. We can see that our HCP always outperforms the two baselines for the points of our HCP are located in the left bottom corner, compared with points of SCP and CCCP. Figure 10 shows the detailed results of CovGap vs. scale and AvgSize vs. scale. For most cases, as the desired error rate becomes smaller, the set size tends to be larger in order to satisfy the coverage guarantee. The results on the LiDAR-based OCC model LMSCNet (Roldao et al. (2020)) show that our HCP is effective in LiDAR-based OCCs, even though they are not the primary focus of our work.

## A.9 Experimental Results on Occ3D-nuScenes Dataset

To demonstrate the scalability of our $\alpha$-OCC approach, we applied it to the Occ3D-nuScenes dataset (Caesar et al. (2020)) and the OCC model BEVStereo (Li et al. (2023c)).

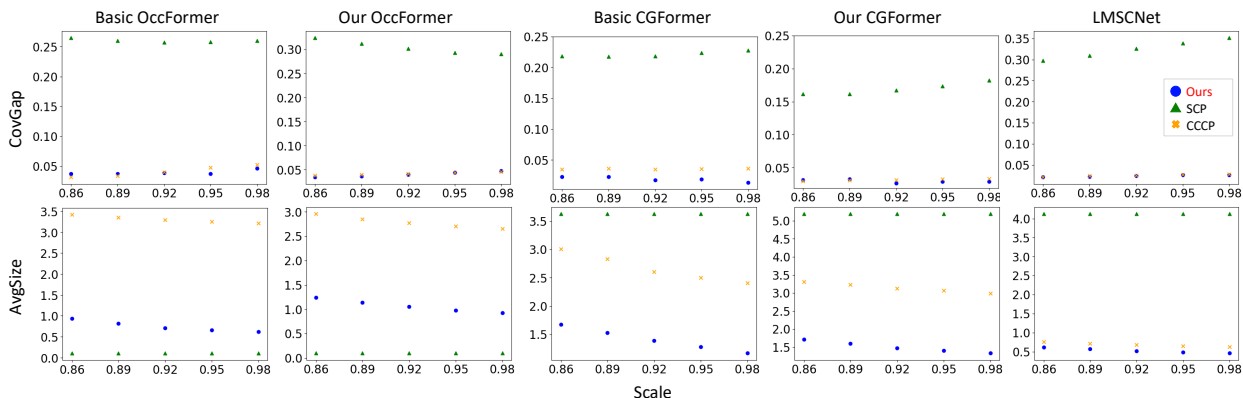

Figure 10: The results of CovGap vs. scale and AvgSize vs. scale for our HCP, SCP and CCCP on SemanticKITTI. The considered OCC models are the basic OccFormer, the OccFormer with our Depth-UP, the basic CGFormer, the CGFormer with our Depth-UP, and the LiDAR-based OCC model LMSCNet. The scale represents the desired class error rate.

The Occ3D-nuScenes dataset consists of 1,000 outdoor driving scenes captured using six surround-view cameras. The sparse input to the OCC model comprises six RGB images from these cameras. The dataset features voxel grids labeled with 17 classes (16 semantic classes and 1 empty class) at a resolution of 0.4m × 0.4m × 0.4m. We utilized only the training and validation sets of NuScenes, as the test set annotations are unavailable. The 3D volume of interest covers a range of 40m ahead and behind the vehicle, 40m to the left and right sides, 1m below, and 5.4m above the vehicle. BEVStereo (Li et al. (2023c)) serves as a commonly used OCC baseline for the Occ3D-nuScenes dataset in many works, such as RenderOcc (Pan et al. (2024b)) and PanoOcc (Wang et al. (2024)). Due to time and computational constraints, both the base BEVStereo model and BEVStereo with our Depth-UP were trained for 12 epochs and 20 batch sizes on the server with 4 Tesla V100 GPUs while the original work trained on 32 batch sizes. The input image size here is 416 × 704 while the original work used the 512 × 1408 input image size.

Table 7 presents the mIoU across all classes and the IoU for each individual class for both the base BEVStereo model and BEVStereo enhanced with our Depth-UP on the Occ3D-nuScenes dataset. Depth-UP demonstrates notable improvements over the base OCC model, achieving a 1.61 (8.77%) increase in mIoU. Furthermore, our Depth-UP significantly enhances performance for small, safety-critical classes, including a 9.43 IoU improvement for the motorcycle class and a 1.09 IoU improvement for the person class. These improvements are attributed to the effective integration of uncertainty information from the depth model into the OCC model.

Table 8 compares our HCP method with SCP and CCCP on the Occ3D-nuScenes dataset using the BEVStereo model, similar to Table 2. The results demonstrate that HCP consistently achieves robust empirical class-conditional coverage while generating smaller prediction sets. Compared to SCP, it reduces the set size by up to 87% and the coverage gap by up to 97%. Similarly, compared to CCCP, it achieves reductions of up to 10% set size and 6% coverage gap. These findings are consistent with experimental results on the SemanticKITTI and KITTI360 datasets, further validating the scalability of our approach.

### A.10 Ablation Study on Model Parameter

Due to space limitations, we did not include the changes in model parameters for our Depth-UP approach in Table 3. Here, we provide a discussion. The original model size is 59.98MB. Adding a head for depth uncertainty estimation increases the size to 60.09MB, representing a 0.18% increase. The Propagation on Geometry Completion (PGC) does not introduce additional parameters beyond this head. In contrast, Propagation on Semantic Segmentation (PSS) increases the parameter size to 71.53MB, a 19.26% increase, primarily due to the additional ResNet backbone required for depth feature extraction. The whole Depth-UP approach, which includes all the above parts, increases the parameter size to 71.53MB. To address this, we

Table 7: Separate results of our Depth-UP on the Occ3D-nuScenes dataset (Caesar et al. (2020)) and the BEVStereo (Li et al. (2023c)) model.

| Method | mIoU | motorcycle | person | traffic-cone | drive-surf. | other-flat | sidewalk | terrain | vegetation | manmade | barries | bicycle | bus | car | const.-veh. | trailer | truck | others |
|---|---|---|---|---|---|---|---|---|---|---|---|---|---|---|---|---|---|---|
| Base | 18.35 | 0.00 | 6.57 | 0.00 | 50.85 | 24.54 | 28.89 | 21.56 | 16.14 | 16.09 | 29.54 | 0.00 | 33.58 | 36.62 | 10.60 | 11.23 | 25.81 | 0.01 |
| Our | 19.96 | 9.43 | 7.66 | 1.71 | 54.01 | 26.84 | 29.76 | 26.19 | 19.14 | 14.44 | 31.74 | 0.01 | 32.55 | 33.43 | 14.25 | 10.00 | 25.02 | 3.10 |

Table 8: Compare our HCP (referred to as "Ours") with the standard conformal prediction (SCP) and class-conditional conformal prediction (CCCP) on CovGap and AvgSize for the Occ3D-nuScenes dataset and the BEVStereo model.

| Method | | Base | | | Our Depth-UP | |
|---|---|---|---|---|---|---|
| CP | SCP | CCCP | Ours | SCP | CCCP | Ours |
| CovGap ↓ | 0.1931 | 0.0070 | 0.0069 | 0.2058 | 0.0075 | 0.0070 |
| AvgSize ↓ | 0.2481 | 0.0341 | 0.0316 | 0.2585 | 0.0376 | 0.0336 |

plan to explore using a simpler backbone for depth feature extraction in future work to reduce the added parameters.

### A.11 Uncertainty vs. Distance

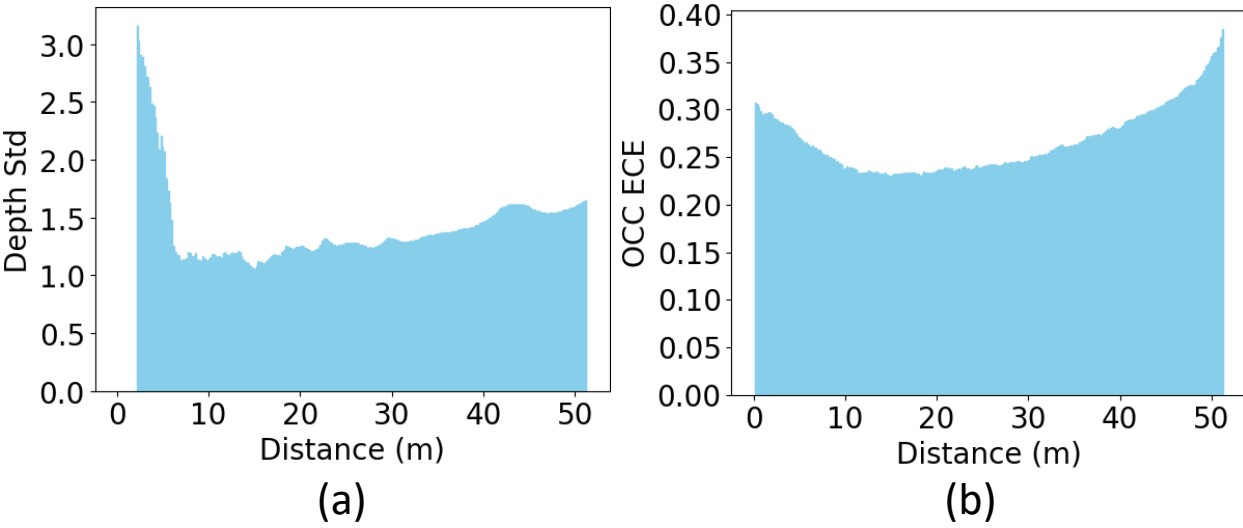

Figure 11: (a) The estimated depth standard deviation from the direct modeling method, representing uncertainty, varies with distance. (b) The Expected Calibration Error (ECE) uncertainty of the VoxFormer model output, representing OCC ECE, varies with distance.

Figure 11 illustrates the relationship between uncertainty and distance in the occupancy prediction model. In Figure 11(a), we show the correlation between the estimated standard deviation (uncertainty) of depth and the distance from the camera to the object. For clarity, we divided the distance into 256 bins, each 0.2 meters in length, and calculated the average estimated standard deviation for each bin. The results reveal that the depth uncertainty is highest when the object is very close to the camera. This phenomenon arises

because, in stereo vision systems, objects at close range result in minimal disparity between the two images, making depth estimation inherently challenging (Wei et al. (2024)). The uncertainty reaches its lowest point at approximately 15 meters, beyond which it increases with distance. This trend aligns with the inverse relationship between depth and disparity, as well as the reduced pixel resolution available for objects further away from the camera (Wei et al. (2024)). These observations confirm that the depth uncertainty estimation in our model is consistent with theoretical expectations.

Figure 11(b) presents the relationship between the Expected Calibration Error (ECE) metric of the VoxFormer model and the distance to the voxels. ECE is a standard metric for assessing the calibration of uncertainty estimates in probabilistic models (Feng et al. (2021)). In this case, we applied the voxel-based ECE computation method described in Cao et al. (2024). The results show that the OCC uncertainty is minimized at approximately 15 meters, consistent with the depth uncertainty trend observed in Figure 11(a). When voxels are very close to the camera, the OCC ECE is relatively high, likely due to depth estimation errors. Similarly, when voxels are far from the camera, the OCC ECE increases, attributed to the limited pixel resolution for distant objects.

Notably, the similarity in the shapes of the curves in Figure 11(a) and 11(b) highlights the significant influence of depth uncertainty on OCC performance, as discussed in Section 1. These findings reinforce the importance of utilizing the depth uncertainty in improving final OCC performance.

### A.12 Table of Abbreviations

| Abbreviation | Full Term |
|---|---|
| OCC | 3D Semantic Occupancy Prediction |
| HCP | Hierarchical Conformal Prediction |
| Depth-UP | Uncertainty Propagation framework from depth models |
| UQ | Uncertainty Quantification |
| PGC | Propagation on Geometry Completion |
| PSS | Propagation on Semantic Segmentation |
| DM | Direct Modeling |
| mIoU | mean Intersection over Union |
| CovGap | Average Class Coverage Gap |
| AvgSize | Average Set Size |
| SCP | Standard Conformal Prediction |
| CCCP | Class-Conditional Conformal Prediction |

Table 9: Table of Abbreviations and Definitions

