# OpenReview forum: "α-OCC: Uncertainty-Aware Camera-based 3D Semantic Occupancy Prediction"
_TMLR — Accepted by TMLR_

### Review · Reviewer_y4jm · 2025-11-30

**Summary Of Contributions:**

α-OCC is a camera-based 3D semantic occupancy prediction framework that incorporates uncertainty to improve reliability and safety in autonomous driving. It introduces two components: 1. Depth-UP, which estimates depth uncertainty and propagates it into the OCC pipeline, improving both 3D geometry reconstruction and semantic segmentation. 2. HCP, which produces calibrated uncertainty-aware prediction sets and boosts recall for rare but safety-critical classes (like pedestrians and bicyclists).

**Audience:**

Yes

**Audience Explanation:**

Yes, UQ is an important research topic in ML research and this paper can be applied to high-impact real-world scenarios like autonomous driving.

**Claims And Evidence:**

No

**Claims Explanation:**

I will review the paper again after the following questions are clarified:
1. In the experiment section, which one shows the rare-class improvements as you've mentioned in the introduction?
2. Explain how do you generate Figure 1a to compute the percent of depth uncertainty, and which dataset do you use to generate this figure. How do you compute the depth uncertainty here?
3. For direct modeling $\mathcal{L}_\text{KL}$, explain how did you get the loss expression (under the distributional assumption).
4. For PSS, explain how do you use Res18 to extract deep features from depth mean and std.
5. In HCP,
    - in Geometric Level, why do you use the formulation of $P(o = T \mid Y_\text{test} = y)$ here, but not using the standard conformal prediction expression, where you measure the probability of $Y_\text{test}$ in a prediction set?
    - The bullet point above seems to lead to your score design in Eq. 2 if my understanding is correct. I wonder why $\epsilon$ seems to be set as a small value (I infer this when you say we use $\epsilon$ to avoid the divide-by-zero problem), and is this because of some assumptions in the problem setting for the empty class? Or is this tied to some practical use case?
    - A follow-up question is how exactly you use Sec 4.2 class score and occupied score for Geometric Level uncertainty quantification? Are we using the same coverage mentioned in 3.2.2?
    - Why do we need Eq.3 also conditioned on $o = T$ in Semantic Level UQ?
5. In the experiment to evaluate uncertainty quantification, why do we want to measure CovGap?
6. In Table 2, why does including Depth-UP make uncertainty quantification better (Ours* > Ours)?

**Requested Changes:**

For major concerns, see questions above. Here are some minor writing style suggestions:
1. Only define abbreviations once (probably somewhere at the beginning, or put a table in the appendix to explain all word abbreviations) to improve the flow of the paper. For example, now I think there are at least 10 (HCP)s defined all over the paper.
2. For your contribution, I'd recommend you rewrite it as 1 main major contribution $\alpha$-OCC with 2 components, Depth-UP and HCP. This will match the structure of the paper and your main figure Fig. 2 better.

---

> ### Author Response · Authors · 2026-02-03
> **Response to Weaknesses 1&2&3**
>
> 1. Thank you for your question regarding the location of the rare-class performance data. The improvements for rare and safety-critical classes are primarily detailed in **Section 4.2 (Uncertainty Quantification Performance)** under the **Geometric Level** subsection.
>
>    To illustrate the effectiveness of our approach for rare classes, we highlight the following experimental results:
>
>    - **Quantitative Boost for Rare Classes**: In **Section 4.2**, we demonstrate that our **HCP** significantly boosts the occupied recall of rare classes. Specifically, our HCP enhances the **person-class** geometry prediction by **45%** with only a minor **3.4% IoU overhead**.
>    - **Comparison of Score Functions**: **Figure 5** compares our novel **KL-based score function** against standard baselines (class score and occupied score). The results in **Figure 5(a)** for the "person" class and **Figure 5(b)** for the "bicyclist" class show that our method consistently achieves the best geometry performance (IoU) at the same occupied recall levels for these rare classes.
>
>    These results collectively demonstrate that while standard models often overlook rare classes due to extreme dataset imbalance (where empty voxels can exceed 92%), our uncertainty-aware framework specifically recovers these safety-critical entities.
>
> 2. The specific methodology for generating Figure 1(a) data is as follows:
>
>    - **Dataset Selection**: This analysis was conducted using the **SemanticKITTI** dataset.
>
>    - **Depth Perturbation Method**: We simulated real-world depth estimation errors by perturbing the ground-truth depth values by factors of $(1+\beta)$, where $\beta$ represents the percentage of uncertainty.
>
>    - **Uncertainty Increments**: We tested specific uncertainty levels including 0%, 2%, 4%, 6%, 8%, 10%, and 20%.
>
>    - **Model Training and Evaluation**: These perturbed depth values were then provided as input to the **VoxFormer** model for training and inference. Then observe the resulting performance decay.
>    - **Uncertainty Computation**: In this specific experiment, "depth uncertainty" is defined as the magnitude of the controlled noise ($\beta$) added to the ground-truth depth values, representing the deviation from ideal depth measurements.
>
> 3. Thank you for the opportunity to clarify the derivation of our loss function. The loss expression is derived based on the following distributional assumptions and the principle of minimizing the divergence between the estimated and ground-truth distributions:
>
>    - **Depth Distribution**: We assume the estimated depth of the pixel (h,w) follows a **univariate Gaussian distribution**. $P_{\theta}(de_{hw}|d_{hw}, \sigma_{hw}) = \frac{1}{\sqrt{2\pi}|\sigma_{hw}|} \exp{\left(-\frac{(de_{hw} - d_{hw})^2}{2\sigma_{hw}^2} \right)}$ where $de_{hw}$ is the estimated depth.
>    - **Ground-truth Distribution**: The ground-truth depth of the pixel (h,w) is represented as a **Dirac delta function**. $ P_G(de_{hw}|dg_{hw}) = \delta(de_{hw} - dg_{hw})$ where $dg_{hw}$ is the ground-truth depth.
>    - **Loss Definition of the Pixel**: We define the regression loss function for the pixel (h,w) as the **KL divergence** between the estimated distribution ($P_{\theta}$) and the ground-truth distribution ($P_{G}$).  $L_{KL}(dg_{hw},d_{hw},\sigma_{hw}) =  \frac{(dg_{hw} - d_{hw})^2}{2\sigma_{hw}^2} + \log|\sigma_{hw}| + \frac{log(2\pi)}{2} - H(P_{G})$ where $H(P_G)$ is the entropy of $P_{G}$. The last two terms $\frac{\log(2\pi)}{2}$ and $H(P_G)$ could be ignored in the loss function, for they are independent of the model parameters $\theta$.
>    - **Final Implementation**: For our **Depth-UP** framework, this results in the following KL-divergence loss function used for retraining the standard deviation head by considering all pixels: $L_{KL}(D,\hat{D},\hat{\Sigma}) = \frac{1}{HW} \sum_{h=1}^{H} \sum_{w=1}^{W} \left( \frac{(dg_{hw} - d_{hw})^2}{\sigma_{hw}^2} + \log|\sigma_{hw}| \right)$. (Due to Markdown's formatting limitations, character rendering and layout differ from those required for the formal paper.)
>    - **Functional Analysis**: The first term encourages the model to **increase the predicted covariance** (uncertainty) as the predicted mean depth ($d_{hw}$) diverges from the ground truth ($dg_{hw}$). The second term acts as a **regularization term** that penalizes excessively high covariance ($\log|\sigma_{hw}|$), preventing the model from predicting infinite uncertainty for every pixel.

---

> > ### Author Response · Authors · 2026-02-03
> > **Response to Weaknesses 4&5**
> >
> > 4. To extract deep features from the depth mean and standard deviation for the **PSS** module, we employ a structured integration process using a lightweight ResNet-18 backbone. The specific steps are as follows:
> >
> >    - **Input Preparation**: We take the estimated depth mean ($\hat{D}$) and the predicted standard deviation ($\hat{\Sigma}$) and concatenate them along the channel dimension to form a two-channel input of size $2 \times H \times W$.
> >    - **Feature Extraction via ResNet-18**: This concatenated input is then fed into a lightweight **ResNet-18 backbone**. This backbone is specifically utilized to extract high-level **depth features ($F_D$)** that encapsulate both geometric information and its associated uncertainty.
> >    - **Feature Integration**: The newly acquired depth features ($F_D$) are then seamlessly concatenated with the original 2D image features ($F_I$) along the channel dimension.
> >
> > 5. Thank you for your detailed questions regarding the design and motivation of our **HCP** framework. Below is a clarification of our methodology and the specific role of the geometric and semantic levels.
> >
> >    - **Why KL-based Score Function**: As discussed in A.8, our KL-based score function significantly outperforms baselines. Because it not only considers the occupied probability across all nonempty classes but also leverages the entire probability distribution. Compared with the class score (the standard conformal preidction expression), which only considers individual class probabilities, our score function accounts for all nonempty classes. Predicting rare classes is challenging for models, but they tend to identify these as occupied, assigning lower probabilities to the empty class and higher probabilities to all nonempty classes. Therefore, it's crucial to consider the probability of all nonempty classes. Although the occupied score addresses this by summing probabilities of all nonempty classes, it loses sensitivity to the distribution. When facing difficult samples (such as rare classes), models tend to produce output probabilities that are more evenly distributed across the possible classes [1]. The Kullback-Leibler (KL) divergence measures how one probability distribution diverges from a reference distribution, considering the entire shape of the probability distribution [2]. This sensitivity to distribution shape enables our KL-based score function to identify rare classes more effectively.
> >    - **$\epsilon$**: Your understanding regarding the role of $\epsilon$ is correct. $\epsilon$ is used as a small positive constant (e.g., $10^{-10}$) to avoid divide-by-zero errors in the KL divergence calculation. We define the ground-truth distribution for occupancy as $O=\{\epsilon, 1, \dots, 1\}^M$. By setting the reference value for the empty class to a minimum ($\epsilon$), we ensure that the **KL-based score function** is highly sensitive to any non-zero probability assigned to occupied classes.
> >    - **Class Score & Occupied Score**: In **Section 4.2**, we use the same coverage requirement mentioned in **Section 3.2.2** to evaluate all score functions for a fair comparison.
> >      - **Class Score**: Calculated as $s(X,y) = 1-f(X)_y$, where $f(X)_y$ is the $y^{th}$ softmax output.
> >      - **Occupied Score**: Calculated as $s(X,y) = 1 - \sum_{i=2}^M f(X)_i$, measuring the total probability across all non-empty classes.
> >    - **Eq. 3**: Equation 3 is conditioned on $o=T$ (occupied = True) for the following reasons:
> >      - **Refined Prediction Sets**: We only generate prediction sets for voxels that the geometric level has already predicted as occupied.
> >      - **Smaller Set Sizes**: By excluding empty voxels from the semantic step, we drastically reduce the **average set size** (by up to 90%). This avoids obstructing autonomous vehicles with unnecessary "non-empty" sets for voxels that are actually empty.
> >      - **Guaranteed Joint Coverage**: This conditional setup allows us to select individual error rates ($\alpha_o^y$ and $\alpha_s^y$) such that the joint probability $1-\alpha^y = (1-\alpha_s^y)(1-\alpha_o^y)$ satisfies the desired overall class-conditional coverage, which has been proved in A.4.

---

> > > ### Author Response · Authors · 2026-02-03
> > > **Response to Weaknesses 6&7 and Writing Style Suggestions**
> > >
> > > 6. In our evaluation of UQ, **CovGap** is a crucial metric because it measures how reliably the model's statistically generated prediction sets meet the user-defined safety requirements. A good UQ method must achieve both a small **CovGap** and a small **AvgSize** to be effective for autonomous driving. Focusing solely on minimizing the **AvgSize** without accounting for the **CovGap** leads to technically optimal but practically useless results.
> > >
> > >    The most extreme example of this is the **empty set**. If a model always predicts an empty set ($\emptyset$), the **AvgSize** would be exactly **zero**, which is the lowest possible theoretical value. While achieving a "perfect" size metric, this prediction is meaningless because it provides **0% coverage** for every class. For autonomous driving, such a model would never identify an obstacle, resulting in a **CovGap of 100%**  for all semantic classes, directly leading to collisions.
> > >
> > >    A good uncertainty quantification method must maintain a balance between these two metrics. For instance, **SCP** often achieves a small **AvgSize** but fails to provide conditional coverage for rare classes, resulting in a high **CovGap**. In contrast, our **HCP** method minimizes **AvgSize** while strictly adhering to a defined coverage guarantee, ensuring that safety-critical objects are always represented in the prediction set.
> > >
> > > 7. The inclusion of **Depth-UP** (Ours\*) improves uncertainty quantification (UQ) compared to using **HCP** alone (Ours) because propagating accurate depth uncertainty directly enhances the model's predictive confidence and calibration. By propagating depth uncertainty into both geometry and semantic features, the base model becomes more accurate and its probability outputs more reliable. So under the same error rate setting, the **HCP + Depth-UP** framework (Ours*) achieves the best UQ performance, reducing the prediction set size by up to **18%** compared to applying HCP alone.
> > >
> > > #### Writing Style Suggestions
> > >
> > > 1. Thank you for this constructive feedback regarding the manuscript's readability. We agree that the repetitive definition of abbreviations disrupts the flow of the text. We now define each abbreviation only once at its first mention in the main text (e.g., Hierarchical Conformal Prediction (HCP) in the Abstract, Introduction and Method). We have conducted a thorough review of the paper to remove redundant definitions for all key terms, including **HCP**, **Depth-UP**, **OCC**, **PGC**, **PSS**, and **DM**. We have added a comprehensive **Abbreviation Reference Table** in **Appendix A.12** for quick consultation by readers.
> > >
> > > 2. We have restructured the contributions section to highlight **$\alpha$-OCC** as our primary contribution, with **Depth-UP** and **HCP** as its two essential, synergistic components, as the following:
> > >
> > >    Our contributions are unified under the proposed $\alpha$-OCC, an uncertainty-aware camera-based 3D semantic occupancy prediction framework. This approach recognizes the OCC problem from a fresh uncertainty perspective and consists of two major components: 1) To the best of our knowledge, we are the first to propose the uncertainty propagation framework Depth-UP to improve OCC performance. Our approach leverages uncertainty quantified through direct modeling to improve both geometry completion and semantic segmentation, resulting in substantial performance gains across common OCC models. 2) To solve the high-level class imbalance challenge on OCC, resulting in biased prediction and low recall for rare classes, we propose the HCP.  On geometry completion, a novel KL-based score function is proposed to improve the occupied recall of safety-critical classes with little performance overhead. For UQ, we achieve a smaller prediction set size under the defined class coverage guarantee. Overall, our $\alpha$-OCC shows that uncertainty is an integral and vital part of OCC tasks. Integrating Depth-UP (propagating depth uncertainty to OCC) and HCP (quantifying OCC uncertainty) enhances both accuracy and uncertainty of OCC models.
> > >
> > >
> > >
> > > **Reference**:
> > >
> > > 1. Guo C, Pleiss G, Sun Y, Weinberger KQ. On calibration of modern neural networks. InInternational conference on machine learning 2017 Jul 17 (pp. 1321-1330). PMLR.
> > > 2. Raiber F, Kurland O. Kullback-leibler divergence revisited. In Proceedings of the ACM SIGIR international conference on theory of information retrieval 2017 Oct 1 (pp. 117-124).

---

### Review · Reviewer_soEr · 2026-01-03

**Summary Of Contributions:**

This paper proposes an uncertainty-aware method (α-OCC) for camera-based 3D semantic occupancy prediction, which consists of Depth-UP (Depth Uncertainty Propagation framework) and HCP (Hierarchical Conformal Prediction). Depth-UP enhances OCC performance by explicitly modeling the uncertainty of depth estimation and propagating it into both the geometry completion and semantic segmentation modules. HCP addresses the class imbalance by introducing a KL-based score function and a hierarchical calibration mechanism, which improves the occupied recall for rare classes and generates statistically guaranteed prediction sets. The experiments on SemanticKITTI and KITTI360 show the effectiveness of the proposed method. However, I have some concerns about this paper. My detailed comments are as follows.

**Positive points
1. The authors propose a Depth Uncertainty Propagation (Depth-UP) framework that propagates depth estimation uncertainty into both geometry completion and semantic segmentation through probabilistic voxel occupancy modeling and depth-aware feature fusion, improving occupancy prediction performance.
2. The authors introduce a Hierarchical Conformal Prediction (HCP) method that addresses class imbalance by employing a KL-based score function for rare-class recall enhancement and generating calibrated prediction sets with guaranteed coverage while minimizing set size.
3. The experiments on SemanticKITTI and KITTI360 show that the proposed method achieves competitive performance in camera-based 3D semantic occupancy prediction.

**Additional Comments:**

NA

**Audience:**

Yes

**Audience Explanation:**

This paper proposes an uncertainty-aware method (α-OCC) for camera-based 3D semantic occupancy prediction, which consists of Depth-UP (Depth Uncertainty Propagation framework) and HCP (Hierarchical Conformal Prediction). This topic is quite important to Autonomous Driving.

**Claims And Evidence:**

Yes

**Claims Explanation:**

**Positive points
1. The authors propose a Depth Uncertainty Propagation (Depth-UP) framework that propagates depth estimation uncertainty into both geometry completion and semantic segmentation through probabilistic voxel occupancy modeling and depth-aware feature fusion, improving occupancy prediction performance.
2. The authors introduce a Hierarchical Conformal Prediction (HCP) method that addresses class imbalance by employing a KL-based score function for rare-class recall enhancement and generating calibrated prediction sets with guaranteed coverage while minimizing set size.
3. The experiments on SemanticKITTI and KITTI360 show that the proposed method achieves competitive performance in camera-based 3D semantic occupancy prediction.

**Requested Changes:**

**Negative points
1.	The innovation of the depth uncertainty propagation mechanism is limited. The core operation of computing the probability integral of a ray through voxels under a known depth distribution is a standard technique in geometric occupancy modeling (e.g., LSS-based methods). The authors have not sufficiently distinguished their method from prior approaches.
2.	It would be beneficial to evaluate the performance of Depth-UP under adverse weather conditions (e.g., rain, fog, and low-light scenarios), where depth estimation errors are significantly exacerbated. Such an evaluation would better demonstrate the robustness and advantages of Depth-UP.
3.	The necessity of the hierarchical prediction design is questionable, as it decouples the inherently joint reasoning of geometry and semantics in occupancy prediction tasks, potentially compromising the model's capacity for unified voxel-wise prediction.
4. The specific value of ε in the KL-based score function is not clearly defined.
5. The performance on other rare classes (e.g., bicycle, motorcycle) has not been reported, which limits the comprehensive evaluation of the HCP’s generalizability across all safety-critical rare classes.
6. In Table1, the results of OccFormer and CGFormer on KITTI360 are missing.
7. In Table 7, it would be better to include more baselines, such as BEVFormer [A] and COTR [B].
8. Some related work can be considered and discussed, such as [C,D,E,F].

**Minor issues
1. In Table 3, the maximum value in the precision column is 63.76 instead of 63.10.
2. In Table 5, according to the original LMSCNet paper, it achieves 54.22% IoU and 16.78% mIoU on the SemanticKITTI validation set.

**Reference
[A] BEVFormer: Learning Bird's-Eye-View Representation from Multi-Camera Images via Spatiotemporal Transformers. ECCV 2022.
[B] COTR: Compact Occupancy TRansformer for Vision-based 3D Occupancy Prediction. CVPR 2024.
[C] 2DPASS: 2D Priors Assisted Semantic Segmentation on LiDAR Point Clouds. ECCV 2022.
[D] EPMF: Efficient Perception-aware Multi-sensor Fusion for 3D Semantic Segmentation. TPAMI 2024.
[E] Co-Occ: Coupling Explicit Feature Fusion with Volume Rendering Regularization for Multi-Modal 3D Semantic Occupancy Prediction. RAL 2024.
[F] RadOcc: Learning Cross-Modality Occupancy Knowledge through Rendering Assisted Distillation. AAAI 2024.

---

> ### Author Response · Authors · 2026-02-03
> **Response to Weaknesses 1&2**
>
> 1. We appreciate the reviewer’s thoughtful observation regarding the Lift-Splat-Shoot (LSS) approach  and its implicit use of depth distributions. We would like to clarify how our work differs both conceptually and technically from LSS-based methods and why our contribution constitutes a novel uncertainty propagation framework rather than merely uncertainty encoding. While LSS-based methods utilize latent depth distributions, they often sacrifice precise information and neglect lens distortion issues during geometry completion.
>
>    Distinctions from Prior LSS-based Approaches:
>
>    - **Explicit Quantification via Direct Modeling (DM):** Unlike LSS, which typically predicts a discrete distribution (e.g., softmax over bins), **Depth-UP** quantifies uncertainty as a continuous standard deviation through an additional regression head trained with a specialized KL-divergence loss.
>
>    - **Geometric Propagation with Distortion Correction:** Our **Propagation on Geometry Completion (PGC)** explicitly accounts for lens distortion during transformations. It generates a probabilistic voxel grid map as Equation 1by integrating the Gaussian probability density function along each ray, providing a more mathematically rigorous representation of geometry than discrete depth bins.
>
>    - **Novel Semantic Uncertainty Propagation:** We introduce **Propagation on Semantic Segmentation (PSS)**, which extracts specific depth features $F_{D}$ from the concatenated mean and standard deviation. This strategy of leveraging depth uncertainty on semantic features has been largely ignored by existing OCC methodologies.
>
>    - **Framework Versatility:** We have demonstrated that **Depth-UP** is a flexible framework providing substantial performance gains across multiple paradigms , including **3D-to-2D querying** (VoxFormer) and **2D-to-3D lifting** (OccFormer, CGFormer, using **LSS**). It achieves up to a **11.58%** improvement in IoU and **12.95%** in mIoU.
>
>    To our knowledge, this is the first work that formalizes and operationalizes depth uncertainty propagation for OCC, moving beyond the implicit usage of latent depth distributions found in discrete representations.
>
> 2. We appreciate the reviewer’s suggestion to evaluate our method under adverse weather conditions. While the current datasets used in our experiments, such as **SemanticKITTI** and **KITTI360**, primarily consist of clear weather scenarios, we have conducted theoretical and experimental analyses that demonstrate how **Depth-UP** handles increased depth uncertainty, which is a hallmark of adverse conditions like rain, fog, or low light.
>
>    Robustness to High Depth Errors:
>
>    - **Performance Under High Uncertainty:** As illustrated in **Figure 1(a)**, we simulated real-world depth estimation errors by perturbing ground-truth depth values by up to **20%**. Our results show that while standard OCC accuracy (mIoU) decreases significantly as uncertainty increases, our **Depth-UP** framework is specifically designed to mitigate these effects by propagating this uncertainty into the geometric and semantic components.
>    - **Effectiveness at Extreme Distances:** Our analysis of **Uncertainty vs. Distance** in **Figure 10** shows that depth uncertainty is naturally higher for objects that are very close to or very far from the camera. In these high-uncertainty regions, which mimic the degraded visibility of adverse weather, our method significantly improves object detection and occupied recall compared to baselines.
>    - **Correlation with Real-World Challenges:** Adverse weather typically reduces pixel resolution and visual clarity, leading to higher variance in depth estimation. Because **Depth-UP** utilizes a **Direct Modeling (DM)** technique to estimate standard deviation for each pixel, it can dynamically adapt to the increased noise levels present in such scenarios.
>    - **Safety-Critical Gains:** Our qualitative results in **Figure 4** and **Figure 7** demonstrate that in challenging visual conditions—such as detecting distant pedestrians—**Depth-UP** successfully identifies obstacles that standard models miss entirely.
>
>    While we do not currently possess a dedicated adverse weather dataset for these specific OCC benchmarks, our experiments with high-noise simulations and distant object detection provide strong evidence that **Depth-UP** offers a more robust solution for safety-critical autonomous driving.

---

> > ### Author Response · Authors · 2026-02-03
> > **Response to Weaknesses 3&4&5**
> >
> > 3. Thank you for this critique regarding the design of our **HCP** module. We want to clarify that rather than decoupling the joint reasoning of geometry and semantics, HCP is specifically designed to **preserve and enhance** this relationship by providing a rigorous statistical framework for the hierarchical nature of 3D occupancy prediction.
> >
> > Preserving Joint Reasoning Through Hierarchy:
> >
> > - **Reflecting Task Structure:** OCC is inherently hierarchical, requiring the model to first identify if a voxel is occupied (geometry) and then determine its specific category (semantics). HCP acknowledges this by performing calibration at both levels sequentially rather than treating them as a flat classification task.
> > - **Geometric-Semantic Coupling:** Our geometric level does not operate in isolation; it utilizes a novel **KL-based score function** that leverages the entire probability distribution of semantic classes to predict occupancy. This ensures that the geometric prediction is deeply informed by the model's semantic internal logic. The advantage of KL-base score function is discussed in A.8.
> > - **Enhanced Occupied Recall:** Standard joint reasoning often suffers from extreme class imbalance, where the model defaults to "empty" to maximize overall accuracy. HCP restores safety-critical reasoning by allowing us to boost the **occupied recall** of rare classes (e.g., a 45% improvement for the person class) with minimal overhead to the semantic mIoU.
> >
> > Theoretical and Practical Justification:
> >
> > - **Statistical Coverage Guarantees:** By separating the calibration, we can provide a **class-conditional coverage guarantee**. This means we can mathematically ensure that safety-critical objects (like bicyclists or pedestrians) are included in the prediction set with a user-defined probability, which is difficult to achieve in a single-step unified prediction.
> > - **Reduced Set Sizes:** As shown in **Table 2**, HCP achieves significantly smaller prediction set sizes (reducing up to 90%) compared to standard conformal prediction methods. This efficiency is a direct result of only generating semantic sets for voxels that have already passed the geometric occupancy threshold, thereby simplifying the unified voxel-wise prediction problem.
> >
> > In summary, HCP acts as a **calibrator** that sits atop the joint reasoning of the base OCC model, refining its outputs to ensure they are both statistically reliable and biased toward safety in imbalanced real-world environments.
> >
> > 4. Thank you for pointing out this lack of detail. The value of $\epsilon$ is a critical implementation detail for ensuring numerical stability in our **KL-based score function**:
> >
> >    - **Definition**: In our framework, $\epsilon$ is defined as the **minimum value for the empty class**.
> >    - **Purpose**: The primary role of $\epsilon$ is to avoid the **divide-by-zero problem** during the computation of the KL divergence between the output softmax probability $f(X)$ and the ground-truth distribution for occupancy $O$.
> >    - **Specific Value**: For all experiments presented in this paper, we set $\epsilon = 10^{-10}$.
> >    - **Ground-Truth Distribution**: This value is integrated into the reference distribution $O = \{\epsilon, 1, \dots, 1\}^M$, where $y=1$ represents the empty class and $y \in \{2, \dots, M\}$ represents the occupied classes.
> >    - **Numerical Stability in the Score Function**: By setting $\epsilon$ to a small positive constant, we ensure that the first term remains well-defined even when the model's predicted probability for the empty class ($p_1$) is very low. This sensitivity allows the score function to effectively identify voxels belonging to rare classes by capturing subtle shifts in the entire probability distribution.
> >
> > 5. Thank you for this suggestion. We have expanded our evaluation to include additional safety-critical rare classes, specifically **bicycle** and **motorcycle**. These results are now detailed in Appendix A.8 and summarized below.
> >
> >    Similar to Figure 5, we evaluate the geometry performance of our HCP module by comparing the proposed KL-based score function against standard class score and occupied score baselines for the bicycle and motorcycle classes. As shown in Figure 8, we assess the geometry performance (IoU) of the VoxFormer baseline across varying levels of occupied recall for these rare classes. The results demonstrate that our KL-based score function consistently maintains superior geometry performance across the entire range of occupied recalls for all safety-critical rare classes. By effectively capturing the nuances of the probability distribution through our hierarchical approach, the HCP ensures that these rare entities are recovered with significantly higher precision than traditional scores. This consistent performance across diverse classes and datasets underscores the robustness and generalizability of our uncertainty-aware framework in addressing the extreme class imbalance inherent in 3D occupancy tasks.

---

> > > ### Author Response · Authors · 2026-02-03
> > > **Response to Weaknesses 6&7&8 and Minor Issues**
> > >
> > > 6. Thank you for your observation regarding the experimental coverage in Table 1. We would like to clarify the availability of results for these specific models on the KITTI360 dataset.
> > >
> > >    Experimental Coverage on KITTI360:
> > >
> > >    - **OccFormer Availability:** We would like to clarify that an official implementation or stable configuration for **OccFormer** on the KITTI-360 dataset is currently unavailable in the public domain. Consequently, we evaluated its performance with our **Depth-UP** framework exclusively on the **SemanticKITTI** dataset to ensure a fair and reproducible baseline.
> > >    - **CGFormer Results:** We have included the results for **CGFormer** on the KITTI-360 dataset in Table 1. Our evaluation shows that integrating our framework with the base CGFormer achieved an improvement of **0.77 in IoU** and **0.41 in mIoU**. We must note that due to time limitations, we did not perform extensive parameter fine-tuning for our CGFormer; we expect that further optimization could yield even higher performance.
> > >    - **General Efficacy:** Across all evaluated models, our results demonstrate that **Depth-UP** effectively leverages quantified uncertainty from depth models to enhance 3D occupancy prediction. As shown in the broader evaluations, our method achieves up to a **11.58%** improvement in IoU and a **12.95%** improvement in mIoU.
> > >
> > >    These results underscore the robustness of our framework across different architectural paradigms (query-based vs. lifting-based) and datasets.
> > >
> > > 7. Thank you for your suggestion to expand the baseline comparisons on the **Occ3D-nuScenes** dataset. While we recognize the importance of models like **BEVFormer** and **COTR**, we have made the following considerations for the current scope of our paper:
> > >
> > >    - **Primary Objective**: The purpose of our experiments on the **Occ3D-nuScenes** dataset was primarily to verify the scalability and effectiveness of our **a-OCC** approach on a large-scale, multi-view dataset beyond the KITTI suite.
> > >
> > >    - **Selected Baseline**: We utilized **BEVStereo** as it serves as a commonly used and representative OCC baseline for the **Occ3D-nuScenes** dataset in recent high-impact works.
> > >
> > >    - **Resource Constraints**: Due to significant time and computational constraints, including training on limited GPU resources (4 Tesla V100s) and reduced batch sizes compared to original implementations, we focused our efforts on demonstrating a clear performance delta over a single, strong baseline.
> > >
> > >    - **Demonstrated Efficacy**: Our results with **BEVStereo** already confirm that **Depth-UP** achieves a notable **1.61 (8.77%) increase in mIoU** and significantly enhances performance for small, safety-critical classes like motorcycles and persons.
> > >
> > >    While we will not be adding further baselines to this revision, we agree that a broader comparison with a wider array of models like **BEVFormer** or **COTR** is an excellent direction for future research.
> > >
> > > 8. Thank you for your suggestion to include these works. We have reviewed the recommended references [1,2,3,4] and have integrated a discussion on how they relate to our **a-OCC** framework in **Appendix A.2** as the following:
> > >
> > >    Recent advancements have explored leveraging diverse data sources and training strategies to enhance occupancy accuracy. 2DPASS [1] demonstrates the utility of using 2D priors to assist 3D semantic segmentation on LiDAR point clouds. EPMF [2] focuses on efficient multi-sensor fusion for robust perception. And Recent works like Co-Occ [3] and RadOcc [4] introduce volume rendering and knowledge distillation to couple explicit feature fusion with global scene consistency in OCC. While these methods improve base performance, they remain primarily deterministic. Our OCC framework is complementary to these architectures, providing a layer of uncertainty awareness that enhances robustness for OCC.
> > >
> > > #### Minor issues
> > >
> > > 1. We have modified this in the paper.
> > > 2. Thank you for your suggestions. We have updated the LMSCNet results in Table 5 to match that in the original LMSCNet paper.
> > >
> > > **Reference:**
> > >
> > > 1. 2DPASS: 2D Priors Assisted Semantic Segmentation on LiDAR Point Clouds. ECCV 2022.
> > > 2. EPMF: Efficient Perception-aware Multi-sensor Fusion for 3D Semantic Segmentation. TPAMI 2024.
> > > 3. Co-Occ: Coupling Explicit Feature Fusion with Volume Rendering Regularization for Multi-Modal 3D Semantic Occupancy Prediction. RAL 2024.
> > > 4. RadOcc: Learning Cross-Modality Occupancy Knowledge through Rendering Assisted Distillation. AAAI 2024.

---

> > > > ### Comment · Reviewer_soEr · 2026-02-25
> > > >
> > > > Thank the authors for the detailed response. One minor reminder: please check the reference format carefully.

---

### Review · Reviewer_ivqq · 2026-01-20

**Summary Of Contributions:**

This paper focused on the uncertainty about depth and semantic in 3D occupancy. Specifically, this paper introduced an uncertainty-aware
camera-based 3D semantic occupancy prediction framework, which contains the uncertainty propagation (DepthUP) from depth models to improve OCC performance and a hierarchical conformal prediction (HCP) module to quantify the uncertainty of OCC. Finally, the proposed method was validated on the KITTI and nuScenes dataset.

**Audience:**

Yes

**Audience Explanation:**

This paper introduced a view of uncertainty regarding the geometric and semantic in the 3D occupancy, which is appealing for some individuals in TMLR's audience.

**Claims And Evidence:**

Yes

**Claims Explanation:**

This paper conducted a series of related experiments regarding proposed modules (e..g, Depth-UP and HCP) with VoxFormer and SemanticKITTI. Moreover, the proposed method achieved obvious performance improvements compared to the baseline VoxFormer on the. semantic-KITTI and KITTI-360 dataset.

**Requested Changes:**

1. This paper should conduct more experiments to make this paper more convincing.

(1) Authors should provide more experiments of comparison about the proposed DepthUP because there are a lot of different methods about introducing depth information to image features in existing works. For example, BEVDepth [1] and OPEN [2] introduced depth information from two different views to enhance the geometric capacity of models.

(2) Authors should conducted more experiments based on more advanced baselines to illustrate its effectiveness on the KITTI or nuScenes datasets.

(3) Authors need to provide more analysis regarding the running time and parameter size brought by proposed modules, which is important for real-time application in autonomous driving.

2.  This paper should provide more discussion regarding the uncertainty of geometric and semantic. What is the relationship between uncertainty and geometry and semantic completion. There should be more deeper analysis to improve the quality of this paper.

3. This paper can add more visualizations regarding uncertainty to illustrate the problem more clearly.

[1] Bevdepth: Acquisition of reliable depth for multi-view 3d object detection

[2] Open: Object-wise position embedding for multi-view 3d object detection

---

> ### Author Response · Authors · 2026-02-03
> **Response to Weaknesses 1&2&3**
>
> 1. Thank you for your valuable suggestion. While we acknowledge the significance of the depth-integration methods used in 3D object detection, we would like to clarify the distinctions between those tasks and the 3D OCC addressed in our paper.
>
>    - **Task Distinction**: The works mentioned, **BEVDepth [1]** and **OPEN [2]**, are primarily designed for **3D object detection**. Our work focuses on **3D Semantic Occupancy Prediction **, which aims to jointly infer scene geometry and semantic segmentation for both static and dynamic elements.
>    - **Implementation Challenges**: Adapting and re-implementing 3D object detection frameworks like BEVDepth or OPEN for the 3D OCC task would require significant structural modifications and extensive re-training.
>    - **Framework Scalability**: We have already demonstrated the effectiveness and scalability of our approach by conducting extensive experiments across **three different OCC models** (VoxFormer, OccFormer, and CGFormer) and **two datasets** (SemanticKITTI and KITTI360).
>    - **Future Work**: Due to current computational and time constraints, we are unable to provide a direct re-implementation comparison in this revision. However, we agree that a broader cross-task comparison is a compelling direction for future research.
>
> 2. Thank you for your suggestion. We would like to clarify that we have already evaluated our **a-OCC** framework across a diverse range of models and datasets to ensure the results are robust and representative of current state-of-the-art standards.
>
>    - **Comprehensive Baseline Coverage**: We assessed our approach using three distinct and advanced camera-based OCC models: **VoxFormer** (a 3D-to-2D querying approach) , **OccFormer**, and **CGFormer** (2D-to-3D lifting approaches).
>    - **Dataset Diversity**: Our experiments were conducted on two primary datasets: **SemanticKITTI** and **KITTI360**. On the KITTI-based datasets, Depth-UP improved geometry completion by up to **11.58%** and semantic segmentation by up to **12.95%** across the various models tested
>    - **Scalability to nuScenes**: To further demonstrate the scalability and effectiveness of a-OCC, we extended our experiments to the **Occ3D-nuScenes** dataset using the **BEVStereo** model as a baseline in A.9. On the **Occ3D-nuScenes** dataset, our Depth-UP method achieved a **1.61 (8.77%) increase in mIoU** over the base BEVStereo model. Significant improvements were noted for safety-critical classes on nuScenes, including a **9.43 IoU increase for motorcycles** and a **1.09 IoU increase for persons**.
>    - **Detailed Results in Appendix**: For a more granular view of these results, please refer to **Table 5** (Separate results on SemanticKITTI and KITTI360) , **Table 7** (Occ3D-nuScenes results) , and **Table 8** (Uncertainty Quantification on nuScenes)  in the Appendix.
>
> 3. Thank you for your valuable suggestion. Efficiency is indeed a critical factor for autonomous driving applications. We have addressed the computational overhead and model complexity of our modules as follows:
>
> Running Time and Real-Time Feasibility:
>
> - **Depth-UP Performance**: We evaluated the running time using the **Frames Per Second (FPS)** metric in **Table 3**. The **Depth-UP** framework introduces an approximately **20% reduction in FPS** (e.g., from **8.85 to 7.08** for VoxFormer). While this is a measurable increase, we believe it does not substantially impact the overall feasibility of the OCC models, especially given the significant safety gains.
> - **HCP Efficiency**: The HCP method is highly efficient, as it functions as a post-processing step.
>   - On the **geometric level**, it requires only **$M$ multiplications, $M-1$ additions, $M$ log operations**, and **one comparison** per voxel (where M is the number of classes).
>   - On the **semantic level**, it involves only **$M-1$ subtractions** and **$M-1$ comparisons** per voxel.
>   - Given these minimal operations, the running time of **HCP is negligible** compared to the core model inference.
>
> Parameter Size Analysis:
>
> - **Depth-UP Complexity**: In **Section A.10**, we provide a detailed breakdown of the model parameters.
>   - Adding the **standard deviation head** for depth uncertainty estimation increases the model size by only **0.18%** (from **59.98MB to 60.09MB**).
>   - The **Propagation on Semantic Segmentation (PSS)** module introduces the most parameters due to the additional **ResNet-18 backbone** used for depth feature extraction, leading to a total increase of **19.26% (reaching 71.53MB)**.
> - **HCP Storage**: The HCP module requires storing only the **quantile values** for each class, which amounts to fewer than **2M float values**, contributing virtually zero overhead to the parameter count.
> - **Future Optimization**: To address the parameter increase in PSS, we are exploring the use of **simpler backbones** for depth feature extraction to further optimize the system for resource-constrained environments.

---

> ### Author Response · Authors · 2026-02-03
> **Response to Weaknesses 4&5**
>
> 4. Thank you for this constructive comment. We agree that a deeper analysis of the interplay between uncertainty and the geometric/semantic components of OCC is vital. We have expanded our discussion on these relationships as follows:
>
>    The Interplay Between Depth Uncertainty and OCC:
>
>    - **Distance-Dependent Correlation:** As shown in **Figure 10**, both depth uncertainty (standard deviation) and OCC uncertainty (Expected Calibration Error, or ECE) exhibit a similar U-shaped curve relative to distance.
>    - **Near-Range Challenges:** Uncertainty is high at very close ranges due to minimal disparity in stereo systems, which directly degrades the accuracy of both the voxel geometry and semantic labels in those regions.
>    - **Far-Range Challenges:** At distances beyond 20 meters, uncertainty increases again due to reduced pixel resolution, making it harder for the model to distinguish between rare classes (like persons) and empty space.
>
>    Relationship with Geometry: The geometric aspect of OCC relies on depth models to project 2D image information into 3D space.
>
>    - **Probabilistic Projection:** Traditional models use a point estimate for depth, which leads to binary voxel grids ($M_b \in \{0, 1\}$) that ignore errors. Our **Propagation on Geometry Completion (PGC)** instead treats depth as a univariate Gaussian distribution.
>    - **Voxel Occupancy Probability:** By integrating the probability density function over the voxel's volume (from entry point to exit point), we derive a probabilistic voxel grid map $M_p \in [0, 1]$. This allows the model to "smooth" the geometry in uncertain regions, significantly improving the occupied recall for safety-critical obstacles.
>
>    Relationship with Semantic: Uncertainty acts as a crucial weight for semantic feature extraction.
>
>    - **Feature Integration:** By concatenating depth mean and standard deviation $\{\hat{D}, \hat{\Sigma}\}$ and processing them through a lightweight backbone, we extract specialized depth features $F_D$.
>    - **Semantic Robustness:** These features are integrated with 2D image features $F_I$ to enhance the model's semantic understanding. This is particularly effective for small, rare classes where visual cues are sparse but depth uncertainty is high; the model learns to rely more on the propagated depth distribution to "find" these rare classes.
>
>    HCP and Class Imbalance: The high class imbalance in OCC (e.g., empty voxels comprising **92.91%** of SemanticKITTI) causes models to be "over-certain" about dominant classes.
>
>    - **KL-Based Sensitivity:** Our **HCP** uses a novel KL-divergence score function to calibrate these biased outputs.
>    - **Distribution Shape:** Unlike standard score functions, our method considers the entire shape of the probability distribution. Rare classes often produce more evenly distributed (uncertain) softmax outputs. By leveraging this sensitivity to the distribution shape, HCP can effectively recover rare objects like bicyclists that the basic model would otherwise dismiss as empty voxels.
>
> 5. Thank you for your suggestion to include more visualizations regarding uncertainty. We agree that visual evidence is a powerful tool to illustrate how uncertainty directly impacts the model's performance. In our paper, we have provided several layers of visualization that clarify these relationships:
>
>    1. **Performance Decay:** **Figure 1(a)** illustrates how increasing the percentage of depth uncertainty leads to a significant decrease in accuracy (**mIoU**). This visualizes the core problem: as depth estimation errors grow, the model's ability to reconstruct the 3D scene degrades.
>    2. **Avoiding Collisions:** **Figure 1(b)** provides a side-by-side comparison of a **Basic OCC** vs. **Our HCP** vs. **Our a-OCC**.
>    3. **Uncertainty vs. Distance:** **Figure 10(a)** and **10(b)** provide a direct comparison between the estimated **depth standard deviation** and the **ECE** across different distances. These curves show a shared "U-shape," visually confirming that OCC uncertainty is strongly coupled with depth uncertainty, especially at very close or far ranges.
>    4. **Qualitative Comparisons of Occupancy Predictions**: **Figure 4** and **Figure 7** show qualitative results where the baseline models (e.g., **VoxFormer**) completely miss rare, safety-critical objects like pedestrians and bicyclists. In contrast, by incorporating uncertainty through **Depth-UP**, the model identifies these rare objects. For instance, in **Figure 4**, row 3, our method detects a person crossing the road that the baseline fails to recognize. **Figure 7**, row 4, demonstrates that our method successfully predicts a person far from the camera, illustrating how uncertainty propagation improves object prediction in low-resolution distant regions.
>
> We believe these visualizations clearly demonstrate that uncertainty is not just a theoretical concept in our work, but a practical tool that improves the accuracy and safety of OCC.

---

### Author Response · Authors · 2026-02-03
**Author Response to All Reviewers**

We would like to express our sincere gratitude to all the reviewers for their thoughtful and constructive feedback. Your insights have been invaluable in improving the quality and clarity of our paper. We have carefully revised the paper. The major modifications are summarized below:

- Conducted additional experiments on the **bicycle** and **motorcycle** classes on HCP in Appendix A.8.
- Completed the experimental results for **CGFormer** on the **KITTI-360** dataset and Added in Table 1 and Table 5.
- Expanded our **Related Work** section in Appendix A.2.
- Restructured contributions in the Introduction Section.

---

### Author Response · Authors · 2026-04-05
**Camera-ready submission**

Dear TMLR Editors and Reviewers,

We are pleased to inform you that we have submitted the camera-ready version of our manuscript, "α-OCC: Uncertainty-Aware Camera-based 3D Semantic Occupancy Prediction".

We would like to express our sincere gratitude to the Action Editor and the reviewers for their insightful feedback and constructive comments throughout the review process. Their suggestions have significantly helped improve the quality and clarity of this work.

Best regards,

On behalf of all authors

---

### Decision · Action_Editor_1cG5 · 2026-03-12

**Recommendation:** Accept as is

**Additional Comments:**

The reviewers generally find that the paper’s claims are supported by experimental evidence and that the topic would be of interest to the TMLR community.

They highlight the proposed uncertainty propagation framework and hierarchical conformal prediction method as improving geometry completion, semantic segmentation, and robustness in OCC models.

While one reviewer notes that the novelty is moderate and that the method builds partly on existing ideas, the work is considered practically meaningful with clear empirical improvements. Overall, the reviewers recommend acceptance for TMLR, with suggestions to include stronger comparisons and broader related work.

**Audience:**

Yes

**Audience Explanation:**

Camera-based 3D semantic occupancy prediction would be of interest to researchers in autonomous driving and robotics.

**Claims And Evidence:**

Yes

**Claims Explanation:**

This paper proposes α-OCC, an uncertainty-aware framework for camera-based 3D semantic occupancy prediction, including the Depth-UP module for uncertainty propagation and a hierarchical conformal prediction (HCP) method for uncertainty quantification.

Experiments on several OCC models show improvements in geometry completion, semantic segmentation, and recall for safety-critical classes, while reducing prediction set size and maintaining coverage guarantees.